



# Organic aerosol source apportionment in Zurich using extractive electrospray ionization time-of-flight mass spectrometry (EESI-TOF): Part I, biogenic influences and day/night chemistry in summer

Giulia Stefenelli[1], Veronika Pospisilova[1], Felipe D. Lopez-Hilfiker[1,a], Kaspar R. Daellenbach[1,b], Christoph Hüglin[2], Yandong Tong[1], Urs Baltensperger[1], Andre S. H. Prevot[1], Jay G. Slowik[1]

[1] Laboratory of Atmospheric Chemistry, Paul Scherrer Institute (PSI), 5232 Villigen, Switzerland

[2] Empa, Laboratory for Air Pollution and Environmental Technology, 8600 Dübendorf, Switzerland

[a] now at: Tofwerk AG, Uttigenstrasse 22, 3600 Thun

[b] now at: Institute for Atmospheric and Earth System Research / Physics, Faculty of Science, University of Helsinki, Helsinki, Finland

*Correspondence to:* Jay G. Slowik (jay.slowik@psi.ch)

**Abstract**

Improving the understanding of the health and climate impacts of $PM_1$ remains challenging and is restricted by the limitations of current measurement techniques. Detailed investigation of secondary organic aerosol (SOA), which is typically the dominating fraction of the organic aerosol (OA), requires instrumentation capable of real-time, in situ measurements of molecular composition. In this study, we present the first ambient measurements by

a novel extractive electrospray ionization time-of-flight mass spectrometer (EESI-TOF). The EESI-TOF was deployed along with a high resolution time of flight aerosol mass spectrometer (HR-ToF-AMS) during summer 2016 at an urban location (Zurich, Switzerland). Positive matrix factorization (PMF), implemented within the Multilinear Engine (ME-2), was applied to the data from both instruments to quantify the primary and secondary contributions to OA. From the EESI-TOF analysis, a 6-factor solution was selected as the most representative and

interpretable solution for the investigated dataset, including two primary and four secondary factors. The primary factors are dominated by cooking and cigarette smoke signatures while the secondary factors are discriminated according to their daytime (two factors) and nighttime (two factors) chemistry. All four factors showed strong influence by biogenic emissions but exhibited significant day/night differences. Factors dominating during daytime showed predominantly ions characteristic of monoterpene and sesquiterpene oxidation while the

nighttime factors included less oxygenated terpene oxidation products, as well as organonitrates which were likely derived from $NO_3$ radical oxidation of monoterpenes. Overall, the signal measured by the EESI-TOF and AMS showed a good correlation. Further, the two instruments were in excellent agreement in terms of both the mass contribution apportioned to the sum of POA and SOA factors and the total SOA signal. However, while the OOA factors separated by AMS analysis exhibited a flat diurnal pattern, the EESI-TOF factors illustrated significant

chemical variation throughout the day. The captured variability, inaccessible from AMS PMF analysis, was shown to be consistent with the variations in the physiochemical processes influencing chemical composition and SOA



formation. The improved source separation and interpretability of EESI-TOF results suggest it to be a promising approach to source apportionment and atmospheric composition research.

## 1. Introduction

Atmospheric aerosols impact visibility, human health and climate on global scale, therefore a detailed knowledge
of chemical composition, sources and processes is a fundamental prerequisite to develop appropriate mitigation policies. Organic aerosols are an important fraction of the chemical composition and are classified as primary (POA) when directly emitted to the atmosphere and secondary (SOA) when formed in the atmosphere through the oxidation of gas-phase precursors, yielding less volatile products which condense to the particle phase (Hallquist et al. 2009). Overall, organic aerosols account for 20-60% of the total fine particulate mass in the
continental mid-latitudes atmosphere and up to 90% in tropical forested areas (Kanakidou et al., 2005, Carlton et al., 2009). POA emissions typically include combustion of fossil fuels, direct injection of unburnt fuel and lubricants, industrial emissions, plant matter debris, biomass burning, and biogenic emissions (Jacobson et al., 2000). Current models estimate that SOA accounts for a dominant fraction of the total organic particulate mass in the lower troposphere, typically between 50 % (in polluted urban areas) and 90% (Jimenez et al., 2009, Hallquist
et al., 2009, Pye et al., 2010, Spracklen et al. 2011). However our capability to characterize SOA is limited (Heald et al., 2008). SOA precursors can have either biogenic or anthropogenic origins, and although key precursor gases for SOA formation are known, the absolute and relative contributions of different sources to SOA remains challenging to determine. Globally, SOA is dominated by oxidation products of biogenic volatile organic compounds, (including the monoterpene α-pinene, one of the largest sources of secondary biogenic particulate
matter worldwide) resulting in an estimated 90% of SOA from biogenic emissions compared to only 10% from anthropogenic sources (Hallquist et al., 2009). However, it has been shown that interaction between biogenic and anthropogenic volatile organic compounds can significantly enhance SOA production and affect its properties (Weber et al., 2007, De Gouw et al., 2009, Kautzman et al., 2010, Glasius et al., 2011, Hoyle et al., 2011, Emanuelsson et al., 2013, Setyan et al., 2014, Moise et al., 2015). Further, anthropogenic SOA disproportionally
affects regions with higher population and thereby exerts a larger impact on global health than suggested by its global average concentration. Elucidating the sources and physicochemical processes governing SOA concentrations requires measurement techniques with high temporal and chemical resolution, which have proven challenging to achieve. The molecular composition of aerosol particles has so far mostly been investigated offline, using filter or cascade impactor samples which are based on a time-integrating sampling step (typically 1 to 24
hours) followed by post-analysis. This method provides detailed information on individual chemical species and/or functional groups in SOA but can be affected by compositional changes due to adsorption, evaporation, and chemical reactions during sample collection, storage, and/or transfer (Turpin et al., 2000, Hallquist et al., 2009). Further, many sources and processes affecting SOA have characteristic timescales that are too short for this measurement approach. Several online techniques have been developed, which couple thermal desorption
and/or hard ionization with online mass spectrometry. A major advantage of the online techniques over offline techniques is their time resolution (Noziere et al., 2015). For instance, the Aerodyne aerosol mass spectrometer (AMS) and the CHARON-PTR-ToF-MS are both able to describe bulk compositional properties of OA but are subject to significant thermal decomposition and/or ionization-induced fragmentation, especially for molecules of the type found in SOA (Müller et al., 2017). In order to retrieve information at the molecular level while avoiding



ionization-induced fragmentation, a few online semi-continuous measurement techniques using soft ionization or thermal desorption have been developed (e.g. FIGAERO-CIMS, Lopez Hilfiker et al., 2014 and TAG, Williams et al., 2006). These instruments have better chemical resolution and reduced thermal decomposition but still low time resolution. Another important drawback is the segregation of collection and analysis stages, which similarly

to offline techniques open the possibility of reaction on the collection substrate and/or transfer artifacts. Alternatively, a soft ionization called aerosol flowing atmospheric-pressure afterglow (AeroFAPA) is also available. This technique allows mass spectrometric analysis of organic aerosols in real time and it consists of an ion source based on a helium glow discharge at atmospheric pressure. Ionization of the analytes occurs in the afterglow region after thermal desorption and produces mainly intact quasi-molecular ions (Brüggemann et al.,

2015). The method though is best suited for polar analytes with high volatilities and low molecular weights while for compounds with low vapor pressures, containing long carbon chains and/or high molecular weights, desorption and ionization is in direct competition with oxidation of the analytes, leading to the formation of adducts and oxidation products which impede a clear signal assignment in the acquired mass spectra. In addition, the ionization pathways are not well-constrained, which makes non-linear behavior likely. All these limitations (decomposition,

fragmentation, reaction/transfer artifacts) are particularly problematic for SOA species which is the fraction of which sources and reactions are least understood. Therefore, instrumentation is urgently needed that can assess original molecular information of organic aerosol, online, with high time resolution and with a linear response to mass.

Here we present the first field deployment of a recently developed extractive electrospray ionization time-of-flight

mass spectrometer (EESI-TOF) (Lopez-Hilfiker et al., 2019), which to our knowledge is the first instrument capable of online OA measurements at atmospheric concentrations using a controlled ionization scheme without thermal decomposition, ionization-induced fragmentation, or separated collection/analysis stages. The field campaign took place during summer 2016 at an urban background site in Zurich, the largest city in Switzerland; the companion paper presents results from a subsequent winter campaign (Qi et al., 2019). This study compares

EESI-TOF and AMS results in terms of both bulk composition and source apportionment to characterize the EESI-TOF field performance and gain new insight into the sources and physicochemical processes governing OA composition.

## 2. Method

### 2.1 Field campaign

Continuous online measurements were performed between June and July 2016 at the Swiss National Air Pollution Monitoring Network (NABEL) station located in Zürich Kaserne, Switzerland (47°22′42″ N, 8°31′52″ E, 410 m above sea level) (Herich et al., 2011). Zürich has a population of 1.3 million, and the site is located in the central metropolitan area, in a courtyard approximately 500 m south of the main train station. This location is not affected by major emissions from industries, but surrounded by roads with rather low traffic, apartment buildings, small

businesses, and restaurants. The NABEL measurement station includes a number of long-term measurements, including trace gas monitors for nitrogen oxides ($NO_x$) and ozone ($O_3$, Thermo Environmental Instruments 49C, Thermo Electro Crop., Waltham, MA) and meteorological data including temperature, relative humidity, solar radiation, wind speed and direction. For the intensive campaign, a separate trailer was deployed to house an




additional suite of gas and particle instrumentation, including the EESI-TOF and several other mass spectrometers, as described below.

The measurement site has been characterized in previous studies as "urban background" for $PM_{10}$, $PM_{2.5}$, and $PM_1$ and additional air quality parameters (Hueglin et al., 2005, Lanz et al., 2007, Daellenbach et al., 2016). The city of Zurich is a hub for railways, roads, and air traffic, providing a useful to assess different sources of SOA depending on location and seasonality. Lanz et al. (2007) reported the first PMF study on an AMS dataset acquired at this site in summer 2005, and identified six factors, including traffic, wood burning, cooking and a charbroiling related source along with two secondary sources discriminated according to their volatility and degree of oxygenation. Additional studies on summer OA measurements provided further discrimination of sources including primary vs. secondary and fossil vs. non-fossil. Also studies at other sites in Europe demonstrated that during summer carbonaceous aerosols are mainly of biogenic origin, emitted either through primary emissions or gas-phase oxidation products from biogenic volatile organic compounds (BVOCs) (Genberg et al., 2011; Yttri et al., 2011). Biogenic SOA (BSOA) has been shown to dominate over combustion-derived aerosols during summer (Gelencser et al. 2007, Genberg et al., 2011; Yttri et al., 2011).

## 2.2 Instrumentation

Particle composition was measured by a high resolution time-of-flight aerosol mass spectrometer (HR-ToF-AMS) and an extractive electrospray time-of-flight mass spectrometer (EESI-TOF). Organic gases were measured by a proton transfer reaction time-of-flight mass spectrometer (PTR-TOF-MS). A scanning mobility particle sizer (SMPS) measured particle size distributions. Here we focus on particle phase composition and organic aerosol source apportionment via positive matrix factorization (PMF).

### 2.2.1 High Resolution Time-of-Flight Aerosol Mass Spectrometer (HR-ToF-AMS)

The non-refractory particle composition was monitored by a high resolution time-of-flight aerosol mass spectrometer (HR-ToF-AMS, Aerodyne Research Inc.) equipped with a $PM_1$ aerodynamic lens (DeCarlo et al., 2006; Canagaratna et al., 2007). The HR-ToF-AMS was operated with a temporal resolution of 1 minute. Briefly, aerosol particles are continuously sampled through an aerodynamic lens, which focuses the particles into a narrow beam and accelerates them to a velocity inversely related to their vacuum aerodynamic diameter. The beam impacts a heated element (600°C, $10^{-7}$ torr), where the non-refractory components flash vaporize. The resulting gas is ionized by electron impact (EI, 70 eV) and ion mass-to-charge ratios ($m/z$) are analyzed by a time-of-flight mass spectrometer. The instrument was calibrated for ionization efficiency (IE) at the beginning and at the end of the campaign using 400 nm $NH_4NO_3$ particles following a mass-based method. A composition-dependent collection efficiency (CDCE) was used to correct the measured aerosol mass according to the algorithm of Middlebrook et al. (2012). Data analysis was performed in Igor Pro 6.3 (Wave Metrics) using SQUIRREL 1.57 and PIKA 1.16.

The PMF source apportionment technique (Section 2.3) requires as input the time-series of ions from high-resolution mass spectral fitting along with their corresponding uncertainties. In the case of the AMS, the measurement uncertainties considered in the error matrix account for electronic noise, ion-to-ion variability at the detector, and ion counting statistics (Allan et al., 2003). Following the recommendation of Paatero and Hopke



(2003), variables (*m/z*) with low signal-to-noise (SNR < 0.2) were removed, whereas "weak" variables (0.2 < SNR < 2) were down-weighted by a factor of 2. Further, all variables calculated during the AMS data analysis as a constant fraction of *m/z* 44 ($CO_2^+$), i.e. the OA contributions to $O^+$, $OH^+$, $H_2O^+$, and $CO^+$, were excluded from PMF analysis to avoid overweighting $CO_2^+$. The contributions of these ions were recalculated after obtaining a solution and reinserted in the factor profiles presented here; the total factor mass was likewise corrected. Isotopic species were likewise excluded from the PMF solver and rescaled afterwards to their parent ions. The final input matrix contained 281 ions (excluding isotopes and $CO_2$-dependent ions and 285 ions including the $CO_2$-dependent ions) between *m/z* 12 and 120 and 22182 points in time (with steps of 60 s).

### 2.2.2 Extractive electrospray ionization time-of-flight mass spectrometer (EESI-TOF)

The EESI-TOF provides online, near-molecular-level measurements of organic aerosol composition with high time resolution. The system, which is described in detail elsewhere (Lopez-Hilfiker et al., 2019), consists of a recently developed EESI source integrated with a commercial time-of-flight mass spectrometer capable of mass resolution up to ~4000 Th/Th (Tofwerk AG, Thun, Switzerland). Briefly, particles and gases are continuously sampled through a multi-channel extruded carbon denuder which removes most gas-phase species with high efficiency. After the denuder, particles intersect a spray of charged droplets generated by a conventional electrospray probe and soluble components are extracted. The droplets enter the mass spectrometer through a heated stainless steel capillary, wherein the electrospray solvent evaporates and ions are ejected. Although the capillary is heated to 250 °C, the effective temperature experienced by the analyte molecules is much lower due to the short residence time, and no thermal decomposition is observed. The resulting ions are analyzed by a portable high-resolution time of flight mass spectrometer with an atmospheric pressure interface (API-TOF) (Junninen et al., 2010). The electrospray solution was a 50/50 water/methanol mixture doped with 100 ppm NaI, with spectra detected in positive mode. The NaI dopant almost entirely suppresses ionization pathways other than formation of $Na^+$ adducts, yielding a linear response to mass (without significant matrix effects) and simplifying spectral interpretation. The EESI-TOF alternated between direct sampling (8 minutes) and sampling through a particle filter (3 minutes) to provide a measurement of instrument background (including spray); the difference between these two spectra yields the ambient aerosol composition. Data analysis, including high resolution peak fitting, were performed using Tofware version 2.5.7 (Tofwerk AG, Thun, Switzerland).

Overall the EESI-TOF measured for 3 consecutive weeks during summer in Zurich, achieving >85% data coverage. The remaining ~15% loss of data acquisition was due to instrumental issues, e.g. clogged capillary resulting in loss of the signal or dirty solution to substitute. Concentrations of inorganic species were very low (see Fig. S1) and a Nafion diffusion dryer was used to prevent major changes in relative humidity. No ion-dependent relative response factors were applied. The (NaI)$Na^+$ signal, an approximate surrogate for ion source stability, varied by ± 7.3 % across the entire campaign and exhibited no systematic drift (Fig. S2), and no corrections relating to sensitivity drift were applied.

Source apportionment analysis on the EESI data included 507 ions between *m/z* 139 and 401, all of which were detected as adducts with $Na^+$ except for nicotine, which was observed with an extra hydrogen ($C_{10}H_{14}N_2H^+$, *m/z* 163.123), likely due to hydrogen abstraction from water. Because of this unique ionization pathway, its relative sensitivity is less certain and its response to a changing particle matrix is poorly constrained, e.g. non-linear




response to mass is a possibility. However, the good agreement between the PMF factors for the AMS and the EESI, discussed in Section 3.3, suggests that any such non-linearities are not significant. One unidentified ion was also included in the analysis. The final input matrix contained 4436 points in time (with steps of 300 s, re-averaged from original 2 s). The input matrix of data and error were calculated as follows: 1) Raw data with time resolution of 2 s were processed with Tofware, including high resolution peak fitting to generate an initial data matrix including mass spectra from both direct ambient sampling and the filter blank. 2) Filter periods were interpolated to yield an estimated background spectrum during ambient measurements. 3) The estimated background was subtracted from the filter spectrum and the resulting difference matrix re-averaged to 300 s time resolution for PMF analysis. 4) Ions whose signal was dominated by spray and/or instrument/gas background as defined by a signal-to-noise ratio (SNR) below 2 were excluded from further analysis. 5) The error matrix was calculated according to Eq. 1 following the model of Allan et al. (2003), which accounts for uncertainties related to the measurements ($\delta_{ij}$) and to the background ($\beta_{ij}$).

$$\sigma_{ij} = \sqrt{{\delta_{ij}}^2 + {\beta_{ij}}^2}$$

(1)

The raw measured signal from the EESI-TOF is acquired in ions per seconds (cps) but throughout the text and figures we report the signal measured by the EESI-TOF in terms of the mass flux of ions to the microchannel plate detector (ag s$^{-1}$), to facilitate interpretation of PMF results and comparison with other instruments, both of which are typically described in terms of mass rather than moles. The mass flux of ions is calculated as follows:

$$M_x = I_x \times (MW_x - MW_{cc})$$

(2)

Where $M_x$ is the mass flux of ions in ag s$^{-1}$ and $x$ represents the measured molecular composition. $I_x$ is the recorded signal (cps) measured by the EESI-TOF. $MW_x$ and $MW_{cc}$ represent the molecular weight of the ion and the charge carrier (e.g. H$^+$, Na$^+$), respectively. Note that this measured mass flux can be related to ambient concentration by the instrument flow rate, EESI extraction/ionization efficiency, declustering probability and ion transmission, where several of these parameters are ion-dependent (Lopez-Hilfiker et al., 2019).

### 2.3 Source apportionment

Source apportionment was performed separately on the organic HR-ToF-AMS and EESI-TOF mass spectral time series using positive matrix factorization (PMF) as implemented by the multilinear engine (ME-2) (Paatero, 1997) and controlled via the interface SoFi (Source Finder, version 6.39; Canonaco et al., 2013) programmed in Igor Pro (Wavemetrics, Inc.). PMF is a bilinear receptor model used to describe measurements (in this case the matrix of organic mass spectra as a function of time) as a linear combination of static factor profiles (i.e. characteristic mass spectra), corresponding to specific emission sources and/or atmospheric processes, and their time dependent source contributions as shown in the following equation (Paatero and Tapper, 1994):



$$x_{ij} = \sum_{k=1}^{p} g_{ik} \times f_{kj} + e_{ij} \tag{3}$$

Here $x_{ij}$, $g_{ik}$, $f_{kj}$, and $e_{ij}$ are matrix elements of the measurement, factor time series, factor profiles and residual matrices, respectively. The subscript $i$ corresponds to time, $j$ corresponds to $m/z$, and $k$ corresponds to a discrete factor. The number of factors in the PMF solution, $p$, is determined by the user. The factor profiles are static, but their concentrations vary with time. Eq. 3 is solved for **G** and **F** using a least squares algorithm that iteratively minimizes the quantity $Q$, defined as the sum of the square of the uncertainty-weighted residuals $(e_{ij}/\sigma_{ij})$:

$$Q = \sum_i \sum_j \left( \frac{e_{ij}}{\sigma_{ij}} \right)^2 \tag{4}$$

Whereas PMF does not require any *a priori* assumption regarding sources, ME-2 (Paatero et al. 1999) enables the inclusion of external data and/or constraints in the PMF model to improve factor resolution and uncertainty analysis. This allows for intelligent rotational control of the retrieved solution. That is, because different combinations of **G** and **F** can yield solutions with similar mathematical quality constraining one or more factor profiles can direct the model towards environmentally reasonable, optimally unmixed solutions. The first application of constrained profiles to AMS data was performed by Lanz et al. (2008) and demonstrated improved model performance by resolving spectrally or temporally similar sources not well-separated by conventional PMF. Here constraints are applied by requiring one or more factor profiles to fall within a predetermined range defined by a combination of a reference profile and a scalar ($\alpha$) determining the tightness of constraint. The $\alpha$ value ($0 \leq a \leq 1$) determines the extent to which the resolved factors ($f_{j,solution}$) and ($g_{i,solution}$) may deviate from input values ($f_j$, $g_i$). The following conditions need to be fulfilled:

$$f_{j,solution} = f_j \pm \alpha \times f_j \tag{5}$$

Because of post-PMF renormalization, the actual profile may contain elements that exceed the boundaries defined by Eq. 5. A key consideration for PMF analysis is the number of factors selected by the user. As currently no methodical and completely objective approach exists for choosing the right number of factors, this selection must be evaluated subjectively to provide the most interpretable solution. Factor identification and interpretation likewise require user judgement. Criteria utilized here include investigation of the retrieved factor profiles for distinctive chemical signatures, diurnal cycle characteristics, and correlations between the time series of factors and external measurements. In addition, the evolution of the residual time series as a function of the number of resolved factors is also evaluated (Ulbrich et al., 2009; Canonaco et al., 2013; Crippa et al., 2014).

## 3. Results

### 3.1 AMS PMF



Figure S1 shows time-series of the species concentrations measured by the AMS over the full period of measurements. The organic mass dominates NR-PM1 with a contribution of 74% compared only a 26% contribution from inorganic mass. The total measured organic mass reached a maximum concentration of ~30 µg m$^{-3}$ during the measurement period, with an average concentration of ~3 µg m$^{-3}$. We note evidence of both long-term events and short-term spikes. We selected a five-factor unconstrained PMF solution containing three primary and two secondary factors. The primary factors consisted of hydrocarbon-like organic (HOA) related to traffic, cooking-related organic aerosol (COA) and a cigarette smoke related factor (CS-OA). The secondary factors were separated by their oxygen content, which has been empirically related to volatility (Jimenez et al., 2009), and are classified here as less oxidized-oxygenated organic aerosol (LO-OOA) and more oxidized-oxygenated organic aerosol (MO-OOA) (Zhang et al., 2011). The five-factor solution was preferred because the four-factor solution was not able to separate the HOA and COA factors, while the six-factor solution added an additional OOA factor with a noisy time-series for which no physical interpretation could be found. Higher-order solutions with up to ten factors likewise yielded no additional interpretable factors. Figure 1 shows the mass spectra of the five-factor solution with ions color-coded according to their chemical family ($C_xH_y$, $C_xH_yO_z$, $C_xH_yN_p$, $C_xH_yO_zN_p$, $H_yO_z$ and "other" which includes $C_xO_y$, $C_x$, $O_y$ and sulfur-containing ions). The factor time series and diurnal pattern are shown in the supplement (Fig. S3). The dominant source in mass is LO-OOA, especially during the period with higher temperature, followed by MO-OOA. Specific local events are instead dominated by the primary sources.

The HOA factor is related to fossil fuel combustion, mainly from traffic emissions. These emissions are typically dominated by engine lubricating oil and consist mainly of n-alkanes, branched alkanes, cycloalkanes, and aromatics leading to high signal of the ions $C_nH_{2n+1}^+$ and $C_nH_{2n-1}^+$ (Ng et al., 2011). Prominent contributions of non-oxygenated species at $m/z$ 43 ($C_3H_7^+$), $m/z$ 55 ($C_4H_7^+$) and $m/z$ 57 ($C_4H_9^+$) can be observed. Similar to other studies (e.g., Lanz et al., 2007; Ulbrich et al., 2009; Zhang et al., 2011) HOA exhibits temporal correlations with primary vehicular emissions tracers, such as elemental carbon from traffic ($EC_{tr}$) and $NO_x$ (Zhang et al., 2005).

The COA factor is similar to HOA in that a large fraction of the signal is contributed by $C_xH_y^+$ ions. However, COA has distinctive mass spectral features, typical of the fragmentation of fatty acids. Characteristic peaks include $C_3H_3O^+$ at $m/z$ 55, $C_3H_5O^+$ at $m/z$ 57 and higher molecular weight oxygenated fragments: $C_5H_8O^+$ ($m/z$ 84), $C_6H_{10}O^+$ ($m/z$ 98), and $C_7H_{12}O^+$ ($m/z$ 112). In addition, the COA and HOA factors could be differentiated on the basis of the signal ratio of $C_3H_3O^+$ to $C_3H_5O^+$ as the COA spectrum tends to show a substantially higher $m/z$ 55 to $m/z$ 57 ratio (Mohr et al., 2009; Sun et al., 2011). Reliable molecular tracers of cooking emissions are not typically available, but Fig. S3b shows a diurnal pattern with significant peaks during meal-times, consistent with previous studies.

The CS-OA factor is related to a cigarette smoke signature and the profile is similar to previously reported smoking-related factors measured at the Jungfraujoch (Froehlich et al., 2015) and a German soccer stadium (Faber et al., 2013). Similar to HOA and COA, the profile includes a strong contribution from $C_xH_y^+$, but for CS-OA is shifted towards less saturated ions (branched and n-alkanes, cycloalkanes, and alcohols). Relevant signal can be observed at $m/z$ 41 ($C_3H_5^+$), $m/z$ 43 ($C_3H_7^+$ and $C_2H_3O^+$) and also fragments from aromatic compounds at $m/z$ 77 ($C_6H_5^+$), 91 ($C_7H_7^+$), 105 ($C_8H_9^+$) and 119 ($C_9H_{11}^+$). In addition, this factor is unique in having a significant contribution from $C_5H_{10}N^+$ ($m/z$ 84) which has been attributed to $n$-methyl-pyrrolidine, a tracer for cigarette smoke (Struckmeier et al., 2016). Furthermore, the CS-OA factor exhibits a significantly higher N:C ratio (0.02) compared to the other factors (ranging from 0.003 to 0.01) and explains most of the organic nitrogen signal.



Finally, the OOA factors are characterized by a very high contribution of the signal at $m/z$ 44 ($CO_2^+$), typical of AMS SOA profiles. The LO-OOA spectrum is characterized by prominent peaks at $m/z$ 43 ($C_2H_3O^+$) and $m/z$ 28 ($CO^+$). It resembles LO-OOA factors determined in previous studies at urban sites, as well as newly formed aerosol from forest emissions and biogenic SOA from chamber studies (Zhang et al., 2007a; Lanz et al., 2007; Ulbrich et al., 2009; Hao et al., 2009; Kiendler-Scharr et al., 2009; Ng et al., 2010; Sun et al., 2010; Hao et al., 2014). LO-OOA has an atomic O:C ratio of 0.40 (consistent with the global average of LO-OOA of 0.35±0.14 (Ng et al., 2010)) while the second OOA factor, MO-OOA, is more oxidized with an O:C ratio of 0.50. The mass spectrum of the latter is dominated by $m/z$ 44 ($CO_2^+$) and $m/z$ 28 ($CO^+$). The profile is similar to MO-OOA factors reported at various locations, including urban areas and the boreal forest (Allan et al., 2006; Ulbrich et al., 2009; Sun et al., 2010; Raatikainen et al., 2010; Hao et al., 2014). Overall, LO-OOA includes less oxygenated and possibly freshly oxidized species while MO-OOA includes highly oxygenated species. Furthermore, the LO-OOA/MO-OOA ratio is higher particularly on days with higher OOA concentration, which in turn correspond to sunny weather and warmer temperatures. The strong correlation of this factor with local ambient temperature indicates that LO-OOA is rather locally formed and possibly linked with SOA formed from the oxidation of biogenic emissions (Fig. S5). Similar findings have been reported by Canonaco et al. (2015) at the same site for summer OA measured by an aerosol chemical speciation monitor (ACSM). During summer afternoons, when photochemical processes are most vigorous the formation of SVOOA is enhanced compared to LVOOA formation which typically occurs on a timescale of hours. This is likely due to the formation of semi-volatile oxygenated aerosol produced from biogenic precursor gases, especially monoterpenes, whose emissions increase with ambient temperature.

The diurnal patterns of these two factors are flatter than the POA factors, however the LO-OOA concentrations started increasing from early morning, most likely due to condensation of semi-volatile species and fresh formation of OOA due to photochemistry. Afterwards this factor continuously decreased, possibly due to boundary layer expansion and photochemical conversion to MO-OOA (Fig. 4). However, the LO-OOA concentration remained significantly higher than other primary emissions which suggests that LO-OOA probably forms from the oxidation of primary emissions and/or continued conversion of less oxidized gas phase products into the particle phase. Furthermore, a correlation between concentrations of LO-OOA and nitrate ($NO_3$) was observed ($R$=0.47). Particulate nitrate also represents semi-volatile secondary species, which share similarity with LO-OOA in terms of volatility and its partitioning behavior with temperature while the MO-OOA time series are correlated with sulfate ($SO_4$) (R=0.6) representing a less volatile fraction and suggest that MO-OOA is rather related to longer-lived aged regional SOA (Lanz et al., 2007). Overall, the LO-OOA and MO-OOA components account for 46 % and 25 % of the total organic aerosol mass observed respectively, dominating altogether the total OA concentration. Due to the extent of fragmentation occurring in the AMS system it was not possible to gain any more information about SOA sources apart from their oxygenation/volatility pattern. The inclusion in the analysis of higher detailed chemical composition, provided by the new EESI-TOF system, allowed to distinguish with more detail the SOA processes of formation and oxidation pathways, as outlined in the following.

**3.2 EESI-TOF PMF**

**3.2.1 Selection and overview of the solution**





We present in the following a PMF analysis on the first-ever ambient EESI-TOF data. As discussed in the previous section, PMF analysis of AMS data indicates SOA to be the dominant component but does not provide any direct chemical information indicating the SOA sources. In contrast, PMF analysis of EESI-TOF data yielded several organic aerosol factors related to secondary OA formation. Factors were separated according to different

mass spectral fingerprints and aging processes for a total of six factors including both POA and SOA. This 6-factor solution presented throughout the text is the averaged solution among 795 bootstrap runs, in which a cooking-related factor ($COA_{EESI}$) is constrained using the cleaner cooking-related factor profile retrieved in the 7-factor solution (see Section 3.2.5).

An overview of the factor profiles, time-series and diurnal patterns is presented in Fig. 2, 3 and 4, respectively. Note that the diurnal pattern presented in Fig. 4 refers to the entire measurement period while Fig. S4 shows the diurnal patterns for the same factors calculated for only the overlapping measurement period between EESI-TOF and AMS. We observed two primary factors: cooking-related OA ($COA_{EESI}$) and cigarette smoke-related OA (CS-$OA_{EESI}$). Four SOA factors were resolved; two daytime SOA factors ($DaySOA1_{EESI}$ and $DaySOA2_{EESI}$) and two

nighttime factors ($NightSOA1_{EESI}$ and $NightSOA2_{EESI}$). Each factor is described in further detail in the following sections.

A common criterion used to assess the number of factor selection is the examination of $Q/Q_{exp}$ for an increasing number of solution factors to evaluate the fraction of explained variation in the data. For unconstrained solutions, the $Q/Q_{exp}$ value decreased smoothly from 5.4 to 4.0 as the number of factors increased from two to ten, providing

little insight into the optimal number of factors. The six-factor solution was chosen after constraining the cleaner cooking profile retrieved from the seven-factor solution within the bootstrap analysis (Section 3.2.5). The solution with one factor less provided a mixed primary emissions factor, while the seven-factor solution resulted in an additional non-interpretable splitting of the daytime SOA (as did higher-order solutions).

Figure 4 shows the diurnal patterns of all individual factor as well as of the sum of all four SOA factors from the

EESI-TOF analysis and sum of the two OOA factors from the AMS analysis. While the pattern of the sum of all factors is basically flat suggesting a negligible influence from local sources, each specific secondary factor alone shows quite distinguished diurnal patterns. The diurnal pattern of $DaySOA1_{EESI}$ exhibits a factor of two enhancement in signal between 15:00 and 21:00 while the $DaySOA2_{EESI}$ exhibits the same magnitude of enhancement in signal around 12:00 without a consistent decrease before 1:00. This shift in time between the two

factors could reflect gradients in composition according to lifetime of the compounds, production time, partitioning and reactive environment. Concerning the diurnal pattern $NightSOA1_{EESI}$ peaks during the night between 22:00 and 05:00 while $NightSOA2_{EESI}$ is elevated in the early morning between 04:00 and noon corroborating the shift in chemistry with also day-time oxidants being available.

### 3.2.2 Primary factors ($COA_{EESI}$ and CS-$OA_{EESI}$)

The $COA_{EESI}$ mass spectrum is dominated by long-chain fatty acids and alcohols which are typical of cooking emissions (Liu et al., 2017). For example, $C_{18}H_{32}O_3$ (coronaric acid, $m/z$ 319.2), $C_{18}H_{34}O_2$ (oleic acid, $m/z$ 305.2) and $C_{16}H_{30}O_3$ (2-oxo-tetredecanoic acid, $m/z$ 293.2) are prominent and contribute 2.1%, 1.7%, 1.5%, respectively, to the overall profile signal. The variability of these ions is also dominated by the cooking source. Another prominent peak in the spectrum, accounting for 0.7% of the signal, is $C_6H_{10}O_5$ ($m/z$ 185), which is attributed to





levoglucosan and commonly used as an indicator for primary aerosols originating from biomass combustion (Hennigan et al., 2010; Giannoni et al., 2012) as it is derived from the pyrolysis of cellulose and hemicellulose. The study from Bertrand et al. (in prep) shows $C_6H_{10}O_5$ ($m/z$ 185) to be a very prominent peak in the EESI-TOF mass spectrum of fresh wood burning emissions. Further, the EESI-TOF is probably more sensitive to

levoglucosan than to bulk SOA (Lopez-Hilfiker et al., AMTD. During this study, it is likely emitted from open cooking activities in the vicinity of the measurement site.

The COA$_{EESI}$ and COA$_{AMS}$ factor time-series are well correlated ($R$=0.65) during the overlapping measurement period (20 to 27 June) (Fig. 3), with both showing clear peaks at lunch time and dinner time (Figs. 4 and S4).

The CS-OA$_{EESI}$ mass spectrum is dominated by $C_{10}H_{15}N_2$ (nicotine, $m/z$ 163.12), and $C_6H_{10}O_5$ (levoglucosan)

which contribute 15% and 10%, respectively to the profile signal. Levoglucosan is also a known product of pyrolysis of simple sugars present in tobacco (Talhout et al., 2006). Other prominent signals occur at $m/z$ 197.04 ($C_7H_{10}O_5$), 199.09 ($C_7H_{12}O_5$), 203.1 ($C_6H_{12}O_6$, glucose), 215.05 ($C_7H_{12}O_6$), 227.05 ($C_8H_{12}O_6$), 313.05 ($C_7H_{14}O_{12}$). The CS-OA$_{EESI}$ shows strong correlation with the AMS factors traffic (HOA$_{AMS}$, $R$=0.6) and cigarette smoke (CS-OA$_{AMS}$, $R$=0.73) emission. The correlation further improve when considering the two sources together ($R$=0.77)

suggesting a certain extent of mixing of the two sources within the same factor. The discrimination of a separate factor related solely to traffic was not possible even investigating solutions with a higher number of factors, where only additional non-interpretable secondary sources were discriminated. The inability of the EESI-TOF to resolve a clear traffic-related factor is likely due to the insensitivity of the instrument to the hydrocarbons dominating these emissions (Section 3.3). The diurnal pattern of CS-OA$_{EESI}$ shows a peak during the evening between 21:00

and 23:00, during which the courtyard in which the measurement site is located is typically more crowded. Overall, as expected, the primary factors show low O:C ratios of 0.38 and 0.43, and high H:C ratios of 1.75 and 1.7 for COA$_{EESI}$ and CS-OA$_{EESI}$, respectively.

### 3.2.3 Secondary daytime factors

Two daytime SOA factors (DaySOA1$_{EESI}$ and DaySOA2$_{EESI}$) were resolved from the EESI-TOF PMF analysis (Fig. 2), both of which contain strong signatures of terpene oxidation products.

Prominent monoterpene-derived ions in the DaySOA2$_{EESI}$ factor profile include $m/z$ 239.09 ($C_{10}H_{16}O_5$), 255.08 ($C_{10}H_{16}O_6$) and 271.079 ($C_{10}H_{16}O_7$) while other peaks are tentatively identified as sesquiterpene oxidation products, i.e., $m/z$ 275.16 ($C_{15}H_{24}O_3$), 307.15 ($C_{15}H_{24}O_5$) and 325.162 ($C_{15}H_{26}O_6$). The latter species could also be

dimers from monoterpenes/isoprene oxidation products. However, the urban location of Zurich is under high NO$_x$ regime which limits the formation of dimers, consistent with the absence of signal from C$_{20}$ compounds related to monoterpene dimerization (Yan et al., 2016; Kurten et al., 2016). Thus, we believe that the above mentioned species are likely related to sesquiterpene oxidation products. Overall, the $C_{10}H_{16}O_z$ series accounts for 6.2% of the total profile signal for DaySOA1$_{EESI}$ and 5.3% for DaySOA2$_{EESI}$ (2.5%, 2%, 4.4% and 5.1% for COA$_{EESI}$, CS-

OA$_{EESI}$, NghtSOA1$_{EESI}$ and NightSOA2$_{EESI}$) while the $C_{15}H_{24-28}O_z$ series accounts for 1% of the total profile signal for DaySOA1$_{EESI}$ and 2.3% for DaySOA2$_{EESI}$. Furthermore, other significant series of compounds are present including ($C_9H_{14}O_z$) accounting for 5.8% and 5.2%, ($C_7H_{10}O_z$) accounting for 4.7% and 3.5% and ($C_8H_{12}O_z$) accounting for 6.4% and 5.6% of the total profile signal for DaySOA1$_{EESI}$ and for DaySOA2$_{EESI}$.

The two DaySOA$_{EESI}$ factor spectra are compared in more detail in Fig. 5a, with the carbon number distribution

shown in Fig. 5b. DaySOA1$_{EESI}$ is more shifted towards ions with lower $m/z$ and carbon number. These species



with less than 10 carbon atoms can represent fragmentation products from terpene oxidation either in the gas phase (Molteni et al., 2019) followed by condensation, or during aging in the condensed phase (Pospisilova et al., submitted). However, fragmentation yields in products that are progressively more difficult to distinguish from ring-opening products from the oxidation of aromatic precursors, and we cannot therefore not rule out a

contribution to these ions from aromatic oxidation products. DaySOA2$_{EESI}$ is instead shifted towards higher masses with a carbon atom number typical of sesquiterpene oxidation products and/or dimerization. Overall the two secondary daytime factors show a high apparent O:C ratio of 0.63 and 0.58, and similar apparent H:C ratios of 1.64 and 1.66 for DaySOA1$_{EESI}$ and DaySOA2$_{EESI}$ respectively, consistent with the expected values for biogenic precursors of SOA which exhibit an H:C ratio from 1.2 to 1.7 (Daellenbach et al., 2018) and specifically

monoterpenes and sesquiterpenes with H:C ratio of 1.6.

The two daytime SOA factors do not only exhibit different chemistry but also a different dependency on ambient temperature. Figure 6 shows the correlation of the two daytime SOA factors with the hourly ambient temperature. While DaySOA1$_{EESI}$ does not show a clear dependency on temperature DaySOA2$_{EESI}$ increases exponentially with temperature, consistent with known relationships for terpene emissions and biogenic aerosol in terpene-dominated

regions (Leaitch et al., 2011; Vlachou et al., 2018). This supports the interpretation of DaySOA2$_{EESI}$ as a factor related to local oxidation of biogenic VOCs and DaySOA1$_{EESI}$ as a factor related to more aged or regional air masses. Fig. S5 shows the equivalent relation with temperature for the AMS secondary factors; we note that LO-OOA$_{AMS}$ exhibits an exponential increase with temperature similar to the DaySOA2$_{EESI}$ but with a weaker correlation, suggesting mixing of the two factors identified by the EESI-TOF and possibly also with other sources

not related to biogenic emissions. The time series DaySOA1$_{EESI}$ shows a significant correlation with MO-OOA$_{AMS}$ ($R$=0.54) which typically represents less volatile and more aged/regional, secondary organic aerosol compounds. An even higher correlation is observed between DaySOA2$_{EESI}$ and LO-OOA$_{AMS}$ ($R$=0.91), where LO-OOA$_{AMS}$ is believed to represent semi-volatile and more freshly produced secondary organic aerosol compounds.

### 3.2.4 Secondary nighttime factors

Two nighttime SOA factors (NightSOA1$_{EESI}$ and NightSOA2$_{EESI}$) were resolved from the EESI-TOF PMF analysis (Fig. 2). The differences in composition between the two factor profiles are shown in Fig. 7 where the signal from the two profiles are also summed by carbon number. NightSOA1$_{EESI}$ peaks between midnight and 04:00, decreases to nearly zero shortly after sunrise, and remains near zero until after sunset. Relative to the DaySOA$_{EESI}$ factors, the NightSOA1$_{EESI}$ spectrum includes less oxygenated and more volatile terpene oxidation

products (e.g. $C_{10}H_{16}O_2$ and $C_{10}H_{16}O_3$), which likely partition to the particle phase due to lower nighttime temperatures. In addition, prominent signatures from organonitrates are evident, which are likely derived from nitrate ($NO_3$) radical oxidation of monoterpenes at night. Previous studies in rural areas during summer suggested $NO_3$ oxidation of monoterpenes to contribute a large fraction of the nighttime SOA (Xu et al., 2015; Zhang et al., 2018). Dominant peaks in the spectrum can be observed for the $C_{10}H_{17}O_xN$ species at $m/z$ 286.09 ($C_{10}H_{17}O_7N$),

302.08 ($C_{10}H_{17}O_8N$) and 270.09 ($C_{10}H_{17}O_6N$) which contribute 3.6%, 4.6%, and 2.4%, respectively to the overall profile signal (ag s$^{-1}$) resulting in the highest contributions compared to all the other factors. Another major series of compounds in the spectra is found for $C_{10}H_{15}O_xN$ which can be observed at $m/z$ 268.08 ($C_{10}H_{15}O_6N$), 284.07 ($C_{10}H_{15}O_7N$), 300.07 ($C_{10}H_{15}O_8N$) and 316.06 ($C_{10}H_{15}O_9N$) and contribute 1.6%, 1.5%, 1.8%, and 1.5%, respectively to the profile signal resulting in the highest contributions compared to all the other factors except for





$C_{10}H_{15}O_7N$ and $C_{10}H_{15}O_8N$ which contribute ~1.3% and ~1.7% to the NightSOA2$_{EESI}$ profile signal (ag s$^{-1}$). These species are consistent with NO$_3$ oxidation products of atmospherically relevant monoterpenes such as limonene (Faxon et al., 2018).

The NightSOA2$_{EESI}$ likewise exhibits a strong and consistent diurnal cycle, with a daily maximum at
approximately 09:00, minimum at 21:00, and smooth transitions in between. Like NightSOA1$_{EESI}$, NightSOA2$_{EESI}$ exhibits strong signatures from organonitrates. However, contributions from non-nitrogen-containing species consistent with limonene and α-pinene ozonolysis and phootoxidation are also evident, e.g. $C_9H_{14}O_{5-6}$ and $C_{10}H_{16}O_{4-6}$ (Beateman et al., 2009; Kahnt et al., 2014; Park et al., 2017) as well as species probably consistent with multi-generation terpenes chemistry or aromatic oxidation products suggesting a certain extent of influence
from photochemistry, consistent with the diurnal morning peak of this factor. Dominant compounds in the spectrum can be observed at *m/z* 286.09 ($C_{10}H_{17}O_7N$), 211.058 ($C_8H_{12}O_5$), 225.07 ($C_9H_{14}O_5$), 239.09 ($C_{10}H_{16}O_5$) and 197.042 ($C_7H_{10}O_5$) contributing 1.9%, 1.3%, 1.3%, 1.2%, and 0.9%, respectively to the total profile signal resulting in the highest contributions among all the other factors signal except for the two DaySOA$_{EESI}$ where they contribute with higher percentages between 1.3% and 3% to the profile signal. Overall the two secondary
nighttime factors show similar O:C ratios (~0.6) and H:C ratios (~1.65) while the N:C ratio is higher for NightSOA1$_{EESI}$ (0.46) than for NightSOA2$_{EESI}$ (0.3).

Fig. S5 shows the correlations of the two nighttime SOA factors with ambient temperature. We note that the NightSOA1$_{EESI}$ increases to some extent with temperature, consistent with biogenic aerosol and with the behavior of DaySOA2. The effect is clear for the night points while not visible for the day points which is expected from
the diurnal pattern of the factor going almost to zero during the day. On the other hand, the NightSOA2$_{EESI}$ does not show any clear dependency on the temperature suggesting a combined effect of partitioning, additional chemistry, and possibly additional sources. Overall, during the day there will generally be higher terpene emissions due to higher temperature, but also higher dilution due to an enhanced boundary layer height compared to the night suggesting that, by compensation, terpene-related SOA formation in the lowest layers of the
atmosphere might be similar. As a consequence DaySOA2$_{EESI}$ and NightSOA1$_{EESI}$ might represent first generation chemistry with different oxidants reflecting the availability during the time of the day while DaySOA1$_{EESI}$ and NightSOA2$_{EESI}$ might represent second generation chemistry oxidation products. As an example, some highly functionalized oxidation products from α-pinene photooxidation like $C_{10}H_{16}O_5$ are thought to be second generation oxidation products (McVay et al., 2016). As a consequence of the extensive decomposition and
fragmentation occurring in the AMS system, which particularly affects organonitrates (Farmer et al., 2010), we were not able to resolve any factor related to night chemistry or specific factor dominated by a nitrate signature to compare with the nighttime SOA factors from the EESI-TOF analysis. However, the organonitrate-derived signal in the AMS and that of the EESI-TOF are well correlated. Figure 8 shows the time series of the sum of all $C_xH_yO_zN^+$ ions from the AMS and $[C_xH_yO_zN_p]Na^+$ ions from the EESI-TOF, with $R$=0.7. For the AMS analysis
the MO-OOA$_{AMS}$ is the factor that contributes the most to these nitrogen-containing fragments above mentioned (~50%) followed by the CS-OA$_{AMS}$ (~20%) while for the EESI-TOF analysis the NightSOA1$_{EESI}$ is the major contributor to the nitrogen-containing species (~35%) followed by NightSOA2$_{EESI}$ (~20%).





### 3.2.5 Bootstrap analysis

Bootstrap analysis (Davison and Hinkley, 1997) was conducted to determine the statistical stability and uncertainties of the EESI-TOF PMF solution, evaluate some trends in specific ions, and extent to which factors are discrete versus basis vectors describing compositional gradients. Bootstrap analyses generate a set of new

input data and error matrices for analysis from random resampling of the original input data. This resampling perturbs the input data by randomly choosing rows (time points) of the original matrix which are present several times, while other rows are removed (Paatero et al., 2014); the overall dimensions of the data matrix is kept constant for each resampling. The resampled data made up on average ~64% of the total original data per bootstrap run. We performed 1000 bootstrap runs for a 6-factor solution with all factors unconstrained except for $COA_{EESI}$,

which as discussed above was constrained using the cooking-related spectrum obtained from the 7-factor unconstrained solution. The cleaner spectra and higher correlation with AMS cooking factor ($COA_{AMS}$) compared to the cooking profile discriminated in the unconstrained 6-factor solution, where a clear mixing with other profiles was still present and an additional not meaningful SOA profile was present resulting from a splitting of the SOA factor in the solution with one factor less. The a-value of the constrained $COA_{EESI}$ was randomly selected for each

bootstrap iteration within the range of 0 to 1 with 0.1 step size. Note that each bootstrap run is started from a different initialization point; thus, this methodology also includes the investigation of seed-based variability, accounting for the possibility of local minima in the solution space.

A particular point of interest in the bootstrap analysis was the extent to which (day and/or night) SOA factors mix with each other. Thus, it is important to characterize solutions where factors are distinct or mixed, and in the case

of mixing, to characterize the type of mixing (i.e. which factors are mixed). For this purpose, we adapted the method of Vlachou et al. (2018). The key steps in this method are as follows: (1) creation of a 6-factor base case: this was synthesized from the unconstrained 7-factor solution described above to optimize $COA_{EESI}$, with the split SOA mathematically combined into a single factor (see Fig. S6 for the 7-factor solution); (2) Spearman correlation between the time series and the profiles of each factor from the base case and a bootstrap solution are used to sort

the bootstrap factors, yielding a correlation matrix with the highest correlation values on the diagonal; (3) each correlation coefficient on the matrix diagonal is compared to those on the intersecting row and column to evaluate whether it is the highest by a statistically significant margin (based on a pre-selected significance level $p$ from a $t$-test). Vlachou et al. (2018) rejected any solution failing to meet this criterion; here we retain the solution but classify it as "mixed". For mixed solutions, we then determined which factor(s) were mixed (i.e. which factor(s)

had time series that could not be unambiguously linked to a unique base case factor based on the statistical significance test described above) and classified solutions according to combinations of mixed factors. This allowed a systematic exploration of bootstrapped solutions most likely to have perturbed the boundaries between selected SOA factors.

The analysis of Vlachou et al. (2018) utilized a $p$-value = 0.05; here we conducted a sensitivity test covering $p$-

values ranging from 0.05 to 0.6. For $p$-values lower than 0.3, the only mixing observed was among POA factors (e.g., $p$-value=0.2 yielded mixing between $COA_{EESI}$ and $POA_{EESI}$ for ~100 runs based on time series analysis and ~80 runs based on profile analysis). At a $p$-value of 0.4, showed mixing $NightSOA1_{EESI}$ with $NightSOA2_{EESI}$ and/or $DaySOA2_{EESI}$ for ~50 runs based on time series analysis, while ~ 10 bootstrap runs showed profile mixing between $DaySOA1_{EESI}$ and $DaySOA2_{EESI}$. However, visual analysis of these "mixed SOA" solutions at $p$=0.4

showed solutions where both the factor profiles and time series were not distinguishable from the base case. We




therefore concluded that the SOA factor separation is robust, supporting our treatment of these factors as discrete entities rather than highly interrelated descriptors of composition gradients.

We applied at this point a significance threshold of 0.3 ($p$-value from $t$-test analysis) and extracted all the solutions classified as unmixed. This resulted in 795 accepted solutions out of 1000 runs, with an average a-value of the

constrained COA$_{EESI}$ profile of 0.399. Figure 9 summarizes the averaged extracted solution from the bootstrap analysis, showing the means and standard deviations of these 795 accepted solutions for the diurnal patterns (Fig. 9a) and factor mass spectra standard deviations against relative intensities (Fig. 9b). The uncertainties of the model (which correspond to the standard deviations among retained solutions) are also presented in Fig. 9 and indicate the high stability of the solution. First we calculated the diurnals, then the standard deviation of the mean diurnals

across all bootstrap runs. Thus, the error bars describe variability across solutions (i.e. model uncertainty) and deliberately exclude day-to-day variability in the actual data.

The median percentage uncertainties for the profiles varied between 5.3 and 12% where the highest uncertainties were related to the nighttime SOA factors. The highest diurnal variability was related to CS-OA$_{EESI}$ and DaySOA2$_{EESI}$. Overall, the uncertainties were not of sufficient magnitude to disrupt the diurnal gradients

discussed above or to significantly affect the apportionment of key ions discussed above. This highlights the relatively discrete nature of the factors.

### 3.3 EESI-TOF and AMS comparison

Figure 10a shows the bulk comparison between the EESI-TOF and the AMS total signal for the overlapping

measurement period. The AMS total signal represents the time series of measured organic mass concentration while the EESI-TOF total signal is the sum of the mass fluxes of every detected ion (neglecting Na$^+$ mass, and excluding ions that are high intensity but spray-dominated). Further, no relative sensitivity corrections were applied for the EESI-TOF even though it is known that there is some sensitivity variability (Lopez-Hilfiker et al., 2019).

The results of the two instruments are correlated ($R$=0.81) despite the assumption that all EESI-TOF ions have the same response factor and even though the AMS measured mass includes a primary source related to traffic (HOA$_{AMS}$) that consists mainly of compounds that are insoluble in the electrospray droplets and therefore not visible in the EESI-TOF.

Fig. 10b shows the EESI-TOF signal as a function of AMS mass for the COA and CS-OA primary factors and the

sum of the SOA factors (i.e. total SOA estimated by EESI-TOF and AMS) where SOA is color coded according to the N:C ratios. The AMS and EESI-TOF SOA estimates are highly correlated ($R$=0.90), suggesting that the variability in the composition is well captured by the model and in good agreement between the two instruments. This strong correlation occurs despite significant variation in SOA composition (e.g. enhanced organonitrates at night), suggesting that the differences in relative response factors among different species are not so large as to

significantly bias the overall source apportionment results. However, some differences are apparent. The time of the day is the main driver of SOA composition and the N:C ratio follows a similar pattern. The N:C ratio color coding of SOA shows a generally higher slope for higher N:C ratios. This is likely due to a combination of two factors (1) underestimation of SOA by the AMS due to organonitrate decomposition to the inorganic ions NO$^+$ and NO$_2^+$, which are not included in the calculation of SOA mass; and (2) higher sensitivity of the EESI-TOF to

SOA with a higher nitrogen content.





The cooking factors (COA) and the cigarette smoke factors (CS-OA) retrieved from each instrument are in good agreement with each other as well, although with lower correlation compared to the secondary factors ($R$=0.64 and $R$=0.73, respectively). The AMS/EESI-TOF correlation for CS-OA further suggests that even though nicotine does not ionize by adduct formation with Na$^+$, this alternate pathway does not introduce significant nonlinearities in its detection, at least under the conditions encountered in Zurich during summer. Similar performance was obtained for nicotine detection during winter measurements in Zurich (Qi et al., 2019). Note that the slopes retrieved from the linear correlation in Fig. 10a are proportional to the EESI-TOF mean sensitivity of the compounds comprising each factor. The slope is nearly a factor of 2 higher for SOA than for COA, which may be due to a combination of two factors. First, it is expected that the EESI-TOF may be more sensitive to the highly oxygenated and highly water-soluble components in SOA than to the fatty acids in COA. Second, the AMS relative ionization efficiency for COA has recently been suggested to be approximately two times higher for COA than for bulk organics, due to the higher molecular weight and thermal decomposition characteristics of the molecules comprising COA (Reyes-Villegas et al., 2018). Nevertheless, these correlations indicate that the EESI-TOF signal linearly relates to mass concentration even for complex ambient aerosol, and also suggest that the overall EESI-TOF sensitivity to OA is not subject to significant variation during the study even if the composition dependent relative sensitivities are actually unknown. Therefore, we assume that factor-specific sensitivities are not needed for the interpretation of the EESI PMF solution where the factors describe the variability in composition.

Figures 10c and 10d show the atomic H:C and O:C ratios, respectively for the total SOA as well as the COA and CS-OA factors determined from the EESI-TOF and AMS data. In terms of O:C ratio, the SOA factors show fair consistency with values around 0.6 and 0.5 for the EESI-TOF and AMS analysis, respectively. For the COA and CS-OA factors, the O:C ratio is much lower for the AMS (~0.1) than for the EESI-TOF COA (~0.4). This is again consistent with a reduced sensitivity of the EESI-TOF to hydrocarbon-like molecules due to a lower extraction and/or ionization efficiency. On the other hand the H:C ratios are slightly higher for the EESI-TOF measurements with values of ~1.6, ~1.7, and ~1.8 for SOA, CS-OA, and COA, respectively, compared to ~1.3, ~1.4, and ~1.6, respectively for the AMS analysis. Similar results were also observed for winter aerosol in Zurich (Qi et al., 2019) and for aging experiments of wood burning emissions in an environmental chamber (Bertrand et al., in preparation). This could suggest a reduced sensitivity of the EESI-TOF to low H:C compounds (e.g. aromatic oxidation products) relative to terpene SOA. Alternatively, given that the EESI-TOF sensitivity to laboratory-generated SOA from single-component precursors roughly decreases with decreasing molecular weight (Lopez-Hilfiker et al., 2019), it may be that compounds with a lower H:C ratio occur predominantly in ions with lower carbon number.

It is interesting to note that the O:C ratios calculated from offline AMS source apportionment for a primary cooking related factor (0.10), a traffic related factor (0.06) and a secondary organic factor (0.51) (Bozzetti et al., 2017) are consistent with our AMS analysis (COA$_{AMS}$ 0.1, HOA$_{AMS}$ 0.057 and OOA 0.42-0.5) rather than with the EESI-TOF ratios (COA$_{EESI}$ 0.38 and SOA$_{EESI}$ 0.56-0.62) suggesting that the EESI extraction process (i.e. solubility) alone cannot explain the discrepancies between the two instruments. It is also important to note that the offline method relies on water extraction for 20 min, while the EESI-TOF is a very fast extraction in





water/methanol thus, the solubility dependence is not identical between the offline AMS and EESI-TOF systems. The bulk variabilities of the H:C and O:C ratios for the total EESI-TOF signal vs. that of the AMS are presented in Fig. S7. The trends shown there are consistent with and explained by those of the individual factors as discussed above.

The contribution of each factor from the EESI-TOF PMF analysis over the entire campaign is reported in Fig. 11 along with the total signal measured from the EESI-TOF (ag s$^{-1}$) and the total measured mass from the AMS (µg m$^{-3}$) (top panel). We note that periods with higher signal correspond to periods with higher ambient temperature, above 25 C° (23- 24 June, 1-2 July and 6-7 July). These days are characterized by high contributions from the

SOA factors and when temperature exceeded 30 C° (23-24 June) the contribution of the DaySOA2$_{EESI}$ was higher compared to DaySOA1$_{EESI}$. It has been shown previously that oxidized biogenic VOCs can considerably enhance particulate mass during heat waves (Guenther et al., 1993, Churkina et al., 2017) suggesting a probable relation of the SOA sources discriminated in these analysis with biogenic emissions and especially suggesting a relation between DaySOA2 $_{EESI}$ and oxidation of freshly emitted terpenes from vegetation as previously presented in Fig.

15   6.

The nighttime composition is significantly different, with NightSOA2$_{EESI}$ in particular often being at or above 50% of the total SOA while the AMS analysis does not allow to identify such a factor. This demonstrates the extent to which important chemical variability is missed by the AMS PMF analysis. Figure 12 shows pie charts of the mean EESI-TOF factor contributions over the entire measurements period (Fig. 12a), for only the

measurement period overlapping with the AMS (Fig. 12b) and the mean AMS factor contributions (Fig 12c). We note that the relative contributions of the factors retrieved from the EESI-TOF analysis are consistent for the two measurement periods, with only small variability. This supports the applied approach of comparing the AMS and EESI-TOF PMF solutions for the entire available periods, despite the limited temporal overlap. Overall the primary factors contribute up to ~20% for the EESI-TOF analysis while they reach up to ~ 30% of the total

measured mass for the AMS. The secondary factors on the other hand contribute up to ~80% of the total apportioned signal for the EESI-TOF analysis and ~70% of the total apportioned mass for the AMS.

Between the two instruments, the COA factors exhibit the strongest difference in contribution, with COA$_{AMS}$ accounting for 11.6% of the total measured organic mass while COA$_{EESI}$ for the overlapping period reaches only 5.7%. This could be a consequence of the under-estimation of the relative ionization efficiency (RIE) of COA$_{AMS}$,

discussed earlier, which would result in an overestimation of its measured mass (Reyes-Villegas et al., 2018). Accounting for this effect, e.g. considering a COA RIE of 2 instead of the default 1.4 value, the COA$_{AMS}$ contribution would decrease to 7.2% and as a consequence the HOA$_{AMS}$, CS-OA$_{AMS}$, LO-OOA$_{AMS}$ and MO-OOA$_{AMS}$ contributions would be 7.1%, 11.2%, 48.2% and 26%, respectively improving in this way the agreement with the COA factor extracted from the EESI-TOF analysis. The RIE is although only one of the possible

explanations, another possible reason is that the AMS collection efficiency likely is closer to 1 if cooking aerosols are externally mixed (Middlebrook et al., 2012).

Figure 13 shows the explained variation (EV) of each factor for selected ions in the EESI-TOF dataset, as well as the variation than cannot be explained by the solution. This is a dimensionless quantity that indicates how much each computed factor explained a row (G) or a column (F) of the input data matrix, X. EV values can be interpreted



as the scaled version of the elements of the input matrix, where the loading of each chemical species in each factor is normalized to 1 (Eq. 12, Canonaco et al., 2013).

The compounds explained most by a single factor are nicotine ($C_{10}H_{15}N_2$) of which ~80% of EV is explained by the cigarette factor alone (CS-OA$_{EESI}$), and the fatty acids ($C_{16-18}$ in Fig. 13), of up to 78% of EV is explained by

the cooking factor alone (COA$_{EESI}$). The variability of the nitrogen-containing compounds is mostly explained by the secondary nighttime factors, and with increasing oxygenation, the contributions from the primary factors are drastically reduced. Further, we include in the analysis two series of compounds likely deriving from biogenic emissions ($C_9H_{14}O_x$ and $C_{10}H_{16}O_x$) where EV by the DaySOA$_{EESI}$ factors is higher for the more oxygenated species, while the EV of less oxygenated species is increased for the NightSOA$_{EESI}$ and primary factors. This is

consistent with temperature-driven partitioning, causing the less oxygenated (and thus more volatile) compounds to be depleted in the particle phase during the day. We also included in the analysis two series of compounds that are commonly related to fossil sources ($C_5$-species) and we note the same effect consistent with partitioning described above. Finally, the $C_6H_{10}O_5$ contribution to the total profiles signal is 54% and 45% for the primary and secondary factors, respectively while its variability is almost equally explained by the primary and secondary

factors with similar contributions of 46% and 54%, respectively. This suggests that most likely this chemical formula does not represent exclusively levoglucosan (or other sugars emitted from cellulose pyrolysis) which are enhanced in primary biomass combustion emissions and under summer conditions can be quickly oxidized (Bertrand et al., 2018b). Instead, significant contributions from non-sugar isomers generated by gas-phase oxidation (similar to the rest of the $C_6H_{10}O_x$ series), are likely. For the latter series of compounds the primary

factors contribute to ~30% and the secondary factors ~70% to the total signal while in terms of explained mass weighted variability the series is explained by up to 18% and 81% by primary and secondary factors, respectively. Overall, we were able with the EESI-TOF PMF analysis to separate more SOA factors compared to the AMS analysis where all variability related to secondary components is included in MO-OOA$_{AMS}$ and LO-OOA$_{AMS}$. Further, the DaySOA2$_{EESI}$ and NightOOA1$_{EESI}$ appear related to specific processes (local daytime terpenes

oxidation and local nighttime terpenes oxidation, respectively). On the other hand, the DaySOA1$_{EESI}$ and NightOOA2$_{EESI}$ factors could not be unambiguously related to a single source of gaseous precursors. These latter factors are also more closely related to each other, and likely a convolution of VOC emissions sources and the atmospheric reactions/timescales for conversion to PM.

This result is conceptually similar to PMF analysis of NO$_3$-CIMS measurements of gas-phase highly oxygenated

molecules (HOMs) by Yan et al. (2016) in the Finnish boreal forest during Spring 2012. Several factors were separated and related to different oxidation mechanisms. Overall the most significant separation was observed between daytime and nighttime; the daytime profiles appeared to be dominated by light HOMs and organonitrates derived from monoterpene chemistry initiated by OH reaction in presence of NO while the nighttime profiles appeared to be dominated by HOMs dimers deriving from the oxidation of monoterpenes with O$_3$ and NO$_3$.

Several of the characteristic molecules identified in that analysis are also present in the case study presented here. For example, $C_{10}H_{15}O_8N$ was found to be the major organonitrate representative of daytime HOMs (Kulmala et al., 2013) and in the current study shows the highest contributions from the DaySOA1$_{EESI}$ and NightSOA2$_{EESI}$ factors. Another example is $C_{10}H_{15}O_9N$ which was considered a tracer molecule of daytime processes initiated by O$_3$ reaction there, while in the current study its variability is mostly explained by the less source-specific





DaySOA2$_{EESI}$ and NightSOA1$_{EESI}$ factors. On the other hand, fingerprint molecules related to nighttime chemistry in Finland, e.g. $C_{10}H_{14}O_7$ and $C_{10}H_{14}O_9$, are in the current study mostly explained by NightSOA2$_{EESI}$ and even more so by DaySOA1$_{EESI}$. This suggests that the variability is strongly driven by local source characteristics and environmental conditions, including daylight hours, oxidant concentrations, of oxidant and terpene sinks

variability. Similarly, Zhang et al. (2018) investigated the nature of monoterpene SOA (MTSOA) from FIGAERO-CIMS analysis in a forested area in the southeastern United States influenced by anthropogenic pollution. They found that different chemical processes involving nitrogen oxides ($NO_x$), during day and night, play a central role for the monoterpene SOA produced suggesting a strong anthropogenic–biogenic interaction affecting the ambient aerosol. The diurnal pattern of MTSOA was flat but specifically they found that the majority

of daytime MTSOA was due to fragmentation products of $RO_2+NO$ while during nighttime monoterpenes were most likely oxidized by $NO_3$ which is primarily formed by $NO_2+O_3$. Overall a large fraction of the identified species in the MTSOA are also present in the current study and contributing with different abundance to all the four SOA factors discriminated, suggesting once more the strong biogenic influence of secondary aerosol in summer at the measurement site.

**4. Conclusions**

We present the first field deployment of a novel extractive electrospray ionization time-of-flight mass spectrometer (EESI-TOF), the first instrument capable of near-molecular measurements of organic aerosol (OA) at ambient concentrations using a controlled ionization scheme without thermal decomposition, ionization-induced fragmentation, or separated collection/analysis stages. The EESI-TOF measured for 3 weeks during

summer in Zurich, Switzerland, achieving >85% data coverage without any systematic drift and signal stability within ± 7.3 %. Overall, the campaign demonstrated the EESI-TOF to be a sufficiently robust instrument for field operation.

Positive matrix factorization (PMF) analysis of EESI-TOF mass spectra yielded two primary organic aerosol factors: cooking-related OA (COA$_{EESI}$) characterized by long-chain fatty acids and levoglucosan (likely

influenced by nearby open cooking activities), and cigarette smoke OA (CS-OA$_{EESI}$), with a strong nicotine signature, as well as four secondary factors. The SOA factors were subdivided into two factors enhanced during the day (DaySOA1$_{EESI}$ and DaySOA2$_{EESI}$) and two during night and/or early morning (NightSOA1$_{EESI}$ and NightSOA2$_{EESI}$). All four factors showed strong contributions from ions characteristic of monoterpene oxidation. Signatures consistent with sesquiterpene oxidation products were also observed in the daytime factors. DaySOA2

exhibited a strong exponential relationship with temperature, and the DaySOA1$_{EESI}$ factor mass spectrum was slightly shifted towards ions with fewer carbon atoms. These differences suggest that DaySOA2$_{EESI}$ is more influenced by local oxidation of biogenic emissions, whereas DaySOA1$_{EESI}$ represents more aged aerosol with possible anthropogenic influences from the oxidation of light aromatics. Two secondary nighttime factors were also observed, with one peaking between midnight and 04:00 (NightSOA1$_{EESI}$) and the other (NightSOA2$_{EESI}$)

gradually increasing after sunset to reach a maximum between 07:00 and 09:00. NightSOA1$_{EESI}$ included less oxygenated terpene oxidation products, as well as organonitrates, likely derived from $NO_3$ radical oxidation of monoterpenes. NightSOA2$_{EESI}$ contained the same signatures with somewhat reduced organonitrate content, as well as a stronger contributions from aromatic oxidation products consistent with the onset of photochemistry.



The EESI-TOF analysis was supported and corroborated by the AMS PMF analysis. We observed a good correlation between the total EESI-TOF and AMS organic signals. The apportionment to the sum of POA and SOA factors was very similar in terms of mass contribution and the agreement between the total SOA signals measured by the two instruments was remarkable. However, the diurnal patterns of the SOA factors disclosed a

different picture. While the total sum of the SOA factors exhibited a rather flat diurnal pattern for both instruments, the two AMS OOA factors similarly showed a flat pattern, while the EESI-TOF factors illustrated significant chemical variation throughout the day. The variation in chemical composition described by the EESI-TOF factors was consistent with various physicochemical processes influencing SOA formation, which was not described by the AMS PMF solution. Further, the O:C ratio between the two instruments was correlated but offset, and similarl

differences were found for the H:C ratio. These differences may be due to higher sensitivity of the EESI system for terpene-derived SOA than aromatic-derived SOA, or higher sensitivity to higher molecular weight species (Lopez-Hilfiker et al., 2019). Overall this work highlights the importance of real-time, highly chemically-resolved data, such as that provided by the EESI-TOF, for identification of the key sources and physicochemical processes governing SOA composition, such as the biogenic emission influences and day/night chemistry identified here.

**Authors contribution**

Main author: GS. Experimental work: GS, VP, FRLH, YD. Formal analysis: GS and JS. External data and scientific input: CH, KRD. Supervision: JGS, ASHP,UB.

**Acknowledgements**

This study was funded by the Swiss National Science Foundation (SNSF starting grant BSSGI0_155846). Logistical support by R. Richter (PSI) is gratefully acknowledged.

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



particularly COA and CS-OA.

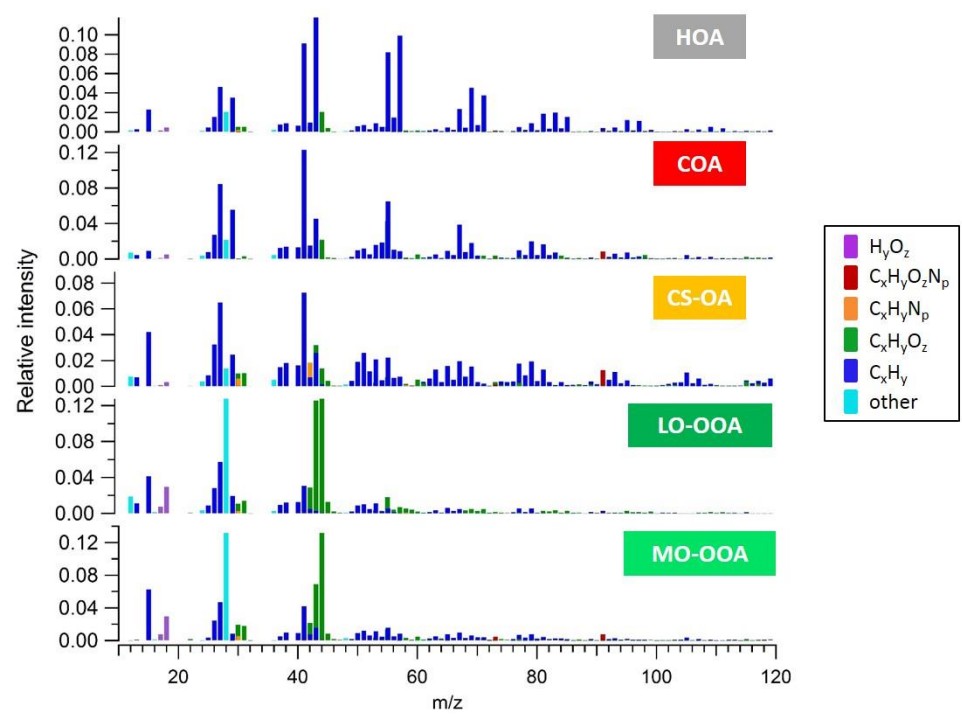

**Figure 1. Mass spectra of the five identified OA factors, color-coded by chemical family.**





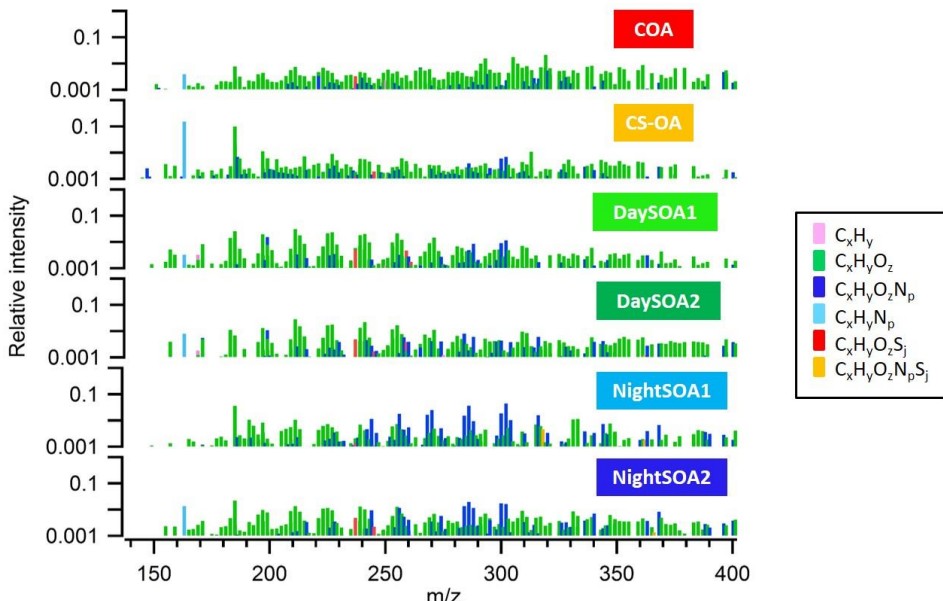

**Figure 2. Mass spectra in log scale of the six identified OA EESI-TOF PMF factors, color-coded according to their chemical families. The sum of each spectrum is normalized to 1.**

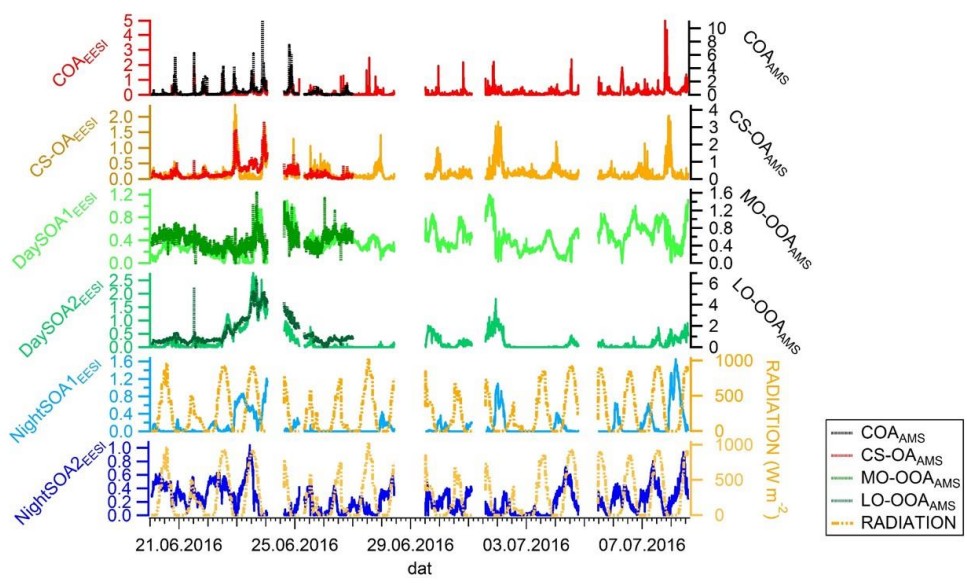

**Figure 3. Time series of EESI-TOF PMF factors (ag s⁻¹) on the left axis and related AMS PMF factors, when applicable, on the right axes (μg m⁻³) for the overlapping measurement period. Solar radiation measurements (W m⁻²) from the**

10    **NABEL station are reported as well as comparison with the night-time EESI factors time-series.**





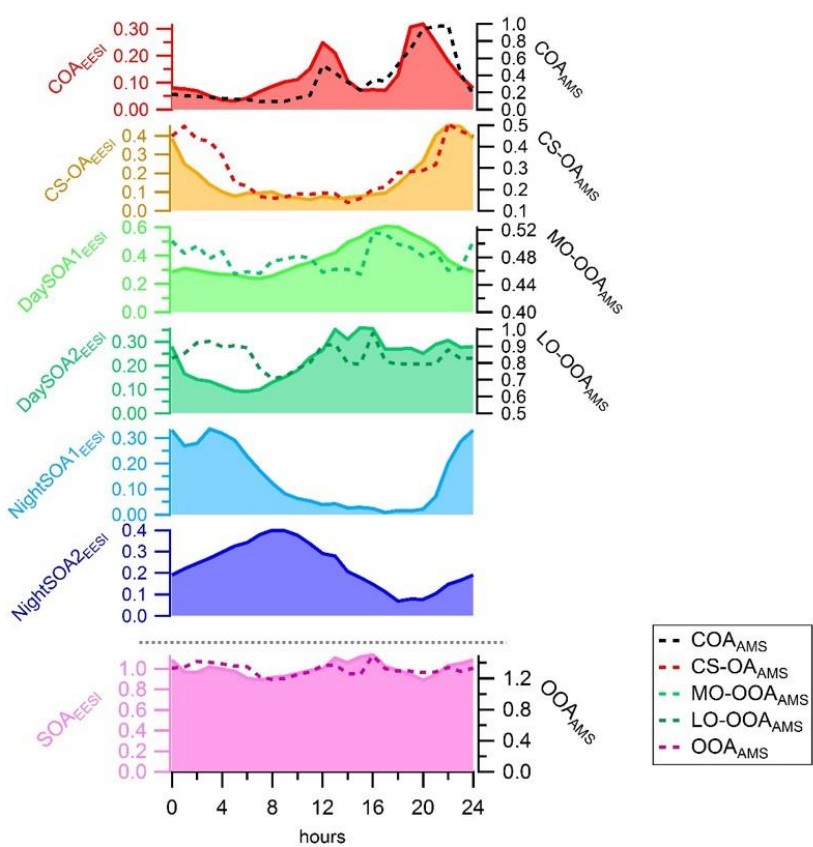

**Figure 4. Diurnal variations of the EESI-TOF PMF factors on the left axis (ag s$^{-1}$) and counterpart diurnal variations from the AMS PMF analysis on the right axis (µg m$^{-3}$). The diurnal variations are here presented for the entire measurement period (see Fig. S4 for the overlapping period only). SOA$_{EESI}$ and OOA$_{AMS}$ denote the sums of all the secondary factors.**





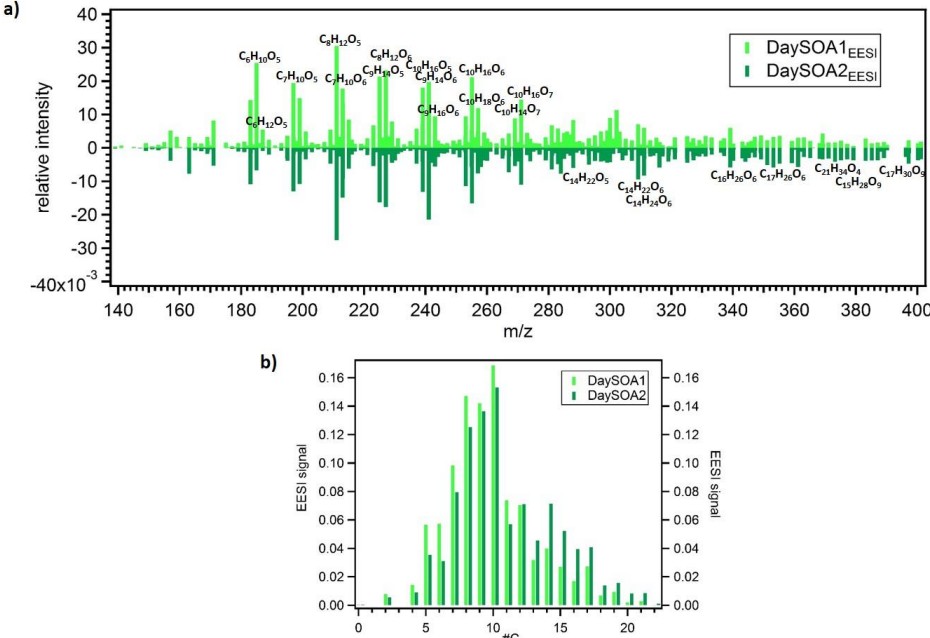

**Figure 5. a) Mirrored mass spectra of EESI DaySOA1 and DaySOA2. Factor profiles are first weighted by their molecular weight to represent equivalent mass concentrations (ag s$^{-1}$) and then normalized such that the sum of each spectrum is 1. b) Histogram of normalized profile signals distributed across bins of carbon atom number.**

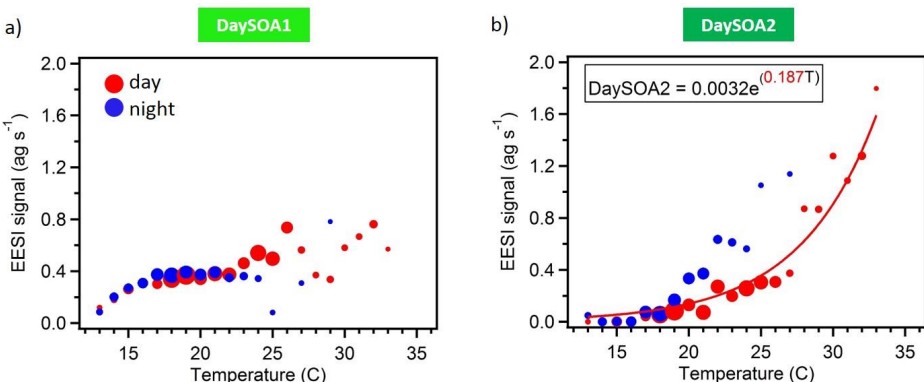

**Figure 6. DaySOA1$_{EESI}$ a) and DaySOA2$_{EESI}$ b) correlation with hourly ambient temperature (C). The data are color-coded according to day (06:00-21:00, red) and night (21:00-06:00, blue) measurements time, they are grouped in temperature bins of 1 C° and the size of the dots correspond to the number of points considered. Data recorded during**

10 **precipitation events is discarded. The fitting curve is weighted by 1/standard deviation.**





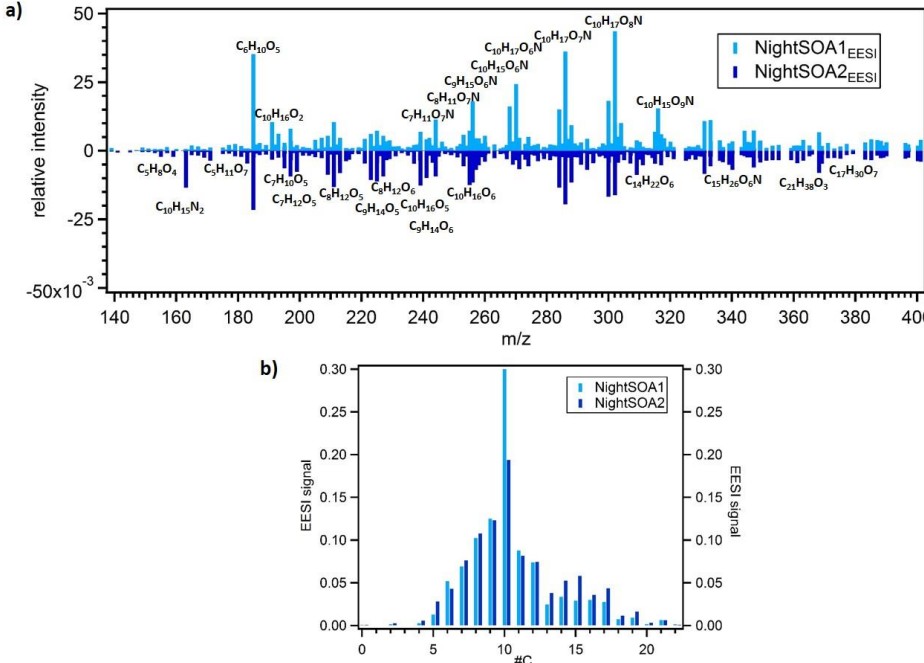

**Figure 7. a) Mirrored mass spectra of EESI NightSOA1 and NightSOA2. Factor profiles are first weighted by their molecular weight to represent equivalent mass concentrations (ag s$^{-1}$) and then normalized such that the sum of each spectrum is 1. b) Histogram of normalized profile signals distributed across bins of carbon atom number.**

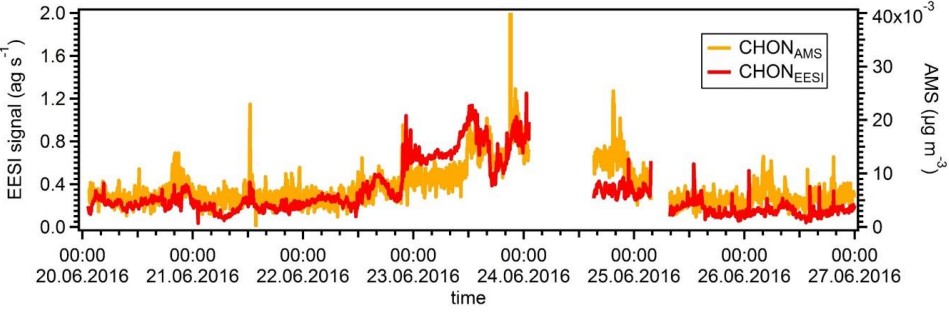

**Figure 8. Time series of the total signal of all $C_xH_yO_zN_p$ species from the AMS (orange trace) and the EESI-TOF (red trace).**




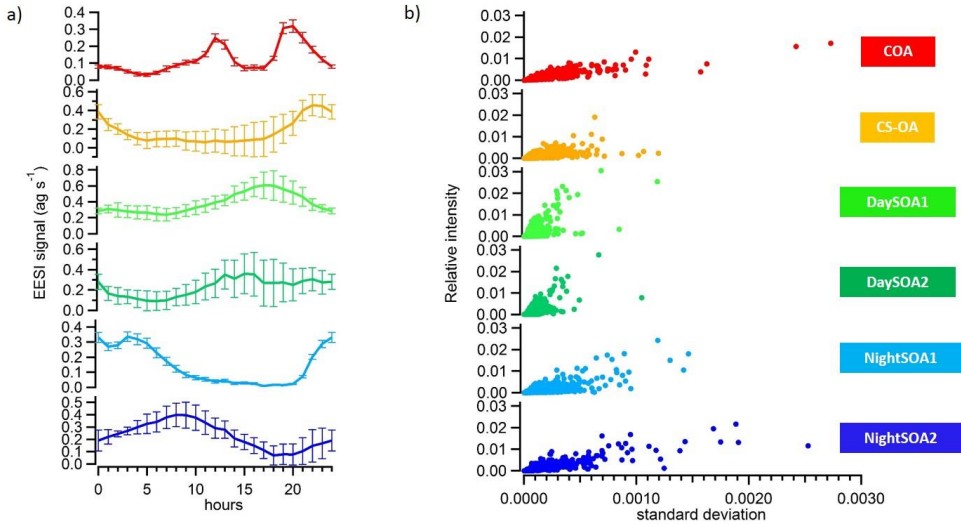

**Figure 9. a) Diurnals variations and b) scatter plots of the relative intensities and standard deviations among 795 bootstrap runs of the six OA factors identified with PMF from the EESI-TOF analysis.**

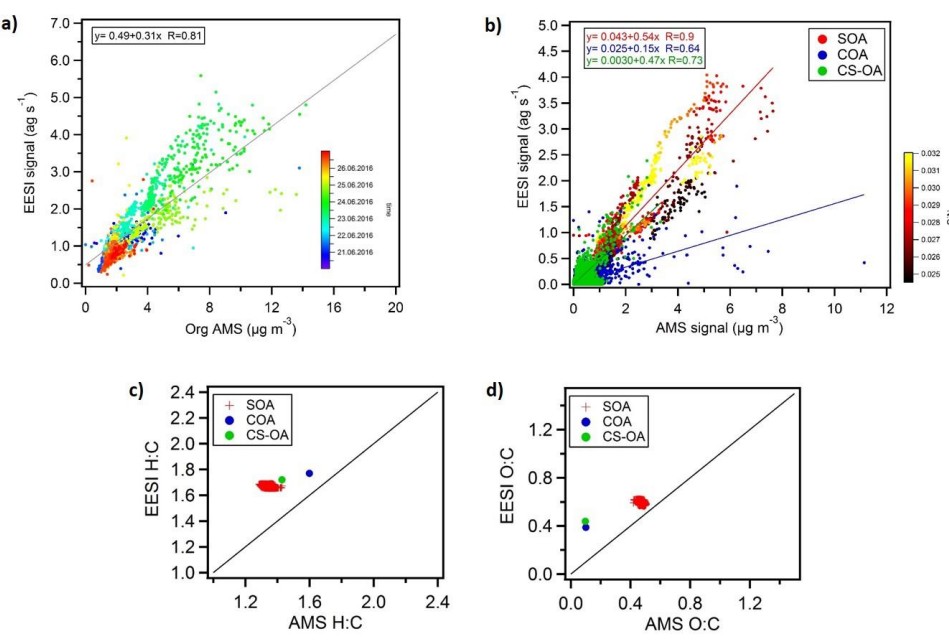



**Figure 10. a)** Total aerosol signal measured by the EESI-TOF versus OA measured by the AMS, with the points colored by time. **b)** Correlations of the CS-OA, COA factors and of SOA (total SOA estimated by EESI-TOF and AMS) from the two instruments, where SOA is color coded by the N:C ratio (yellow dots represent N:C ≥ 0.032) b). **(c), (d)** Atomic ratios, i.e. H:C and O:C ratios for SOA and the CS-OA and COA factors.

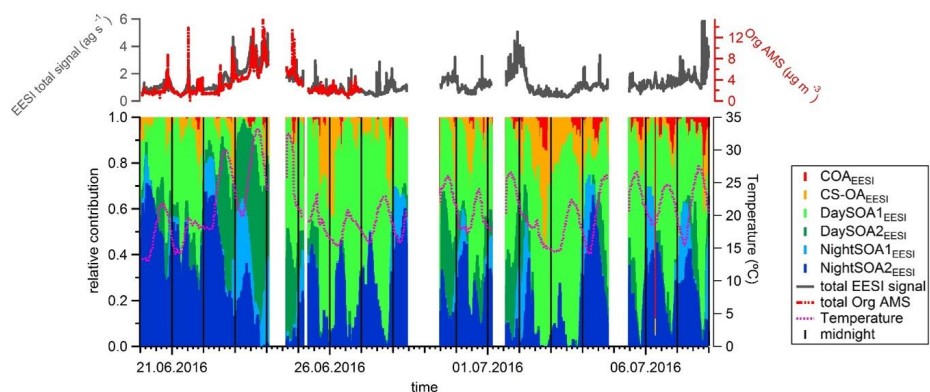

**Figure 11. (Top)** Total signal from the EESI-TOF (ag s⁻¹) and total organics mass concentration measured with the AMS (µg m⁻³). **(Bottom)** Relative contributions of the EESI-TOF factors to the total signal. Vertical black lines denote
10    midnight.

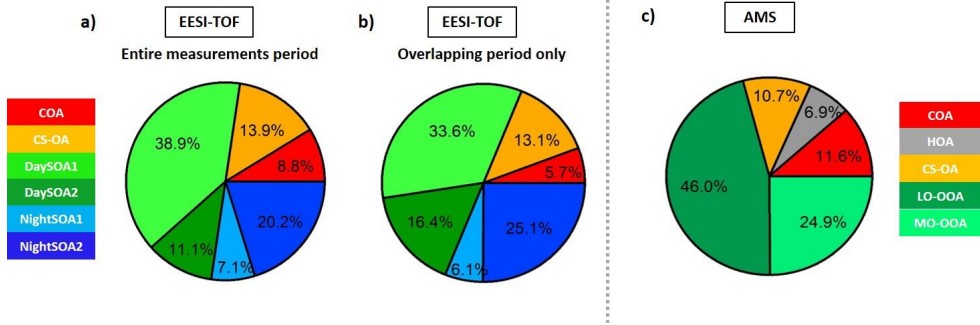

15    **Figure 12. a)** Pie charts of the EESI-TOF factor mean contributions (%) to the total measured signal for a) the entire measurement period and b) the period overlapping with the AMS measurements; c) the AMS factor mean contributions (%) to the total measured organic mass.





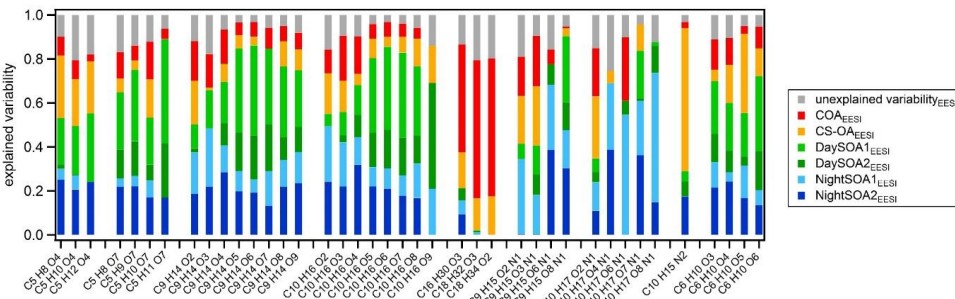

**Figure 13. Explained and unexplained variability for a subset of compounds from the EESI-TOF PMF analysis weighted on the explained variability of each factor. The subset of molecules has been selected according to the interesting species among all the variables in the analysis. Species are ordered according to their chemical composition.**

5 **On average the $C_5$ species contribute 1.7% to the total signal measured, the $C_9H_{14}O_x$ ~4%, $C_{10}H_{16}O_x$ ~4.6%, $C_{16-18}$ ~0.7%, $C_{10}H_{17}O_xN$ ~1.7%, $C_9H_{15}O_xN$ ~1.1%, $C_{10}H_{15}N_2$ ~2.5%, $C_6H_{10}O_5$ ~3.5% and the remaining $C_6H_{10}O_x$ ~0.5%.**

