# Peer review of "Organic aerosol source apportionment in Zurich using extractive electrospray ionization time-of-flight mass spectrometry (EESI-TOF): Part I, biogenic influences and day/night chemistry in summer"

_Atmospheric Chemistry and Physics, 2019_

## Referee Comment (RC1) · Anonymous Referee #1 · 17 Jun 2019

This paper presents the first field observations using the EESI-TOF, a new online ESI-MS instrument for aerosols. Factorisation is applied to the data and is compared against the equivalent factorisation products from an AMS. While the prevalence of cigarette smoke would throw into question the representativeness of the measurement site, it provides an interesting insight into the instrument function. Also, the separation of four different SOA factors adds an additional dimension to the AMS analysis, which typically only breaks it down into two factors.

[Figure]

While this paper has a technical theme, I would consider there to be enough atmospheric science to warrant publication in ACP rather than AMT. It is also very well written and presented. This work will no doubt form the foundation for many future papers using this technique, much as the Lanz et al. papers did for PMF-AMS using data obtained in Zurich. I recommend publication subject to minor comments, however I only have a few of these to make.

Page 5, line 8: It should be specified that the resolution being reported is m/dm. Also, this quantity is normally presented dimensionless; Th/Th isn't really a unit.

Page 10, line 6: Why is the average of the bootstrap runs used as a solution? Bootstrapping is normally used to investigate the robustness of a solution, not to provide the solution itself. While the process is explained in greater detail in section 3.2.5, I still don't see why the average should be considered a more reliable solution than the base case (if it was because of the constraint on the cooking factor, couldn't the optimal a-value simply be applied to the base case?). Regardless, I would have liked to have seen the technical explanation in 3.2.5 given before the presentation of results.

Page 10, line 26: How can you say that there were negligible local influences? The primary factors alone would indicate that local sources were very significant.

Page 12, line 20: Please refrain from using the term 'significant' in a statistical context unless a particular test (e.g. p-test) has been applied.

---

## Referee Comment (RC2) · Anonymous Referee #2 · 2 Aug 2019

This manuscript presents source apportionment from ambient measurements where an extractive electrospray (EESI) mass spectrometer was deployed alongside an aerosol mass spectrometer (AMS) in an urban setting. The paper demonstrates the capability of the EESI to measure variability in ambient organic aerosol that is consistent with that measured by the AMS, and to also identify additional variability in SOA composition that is not isolated in factor analysis of the AMS data.

[Figure]

The comparison of the EESI and AMS total signal and the comparison of the factor analysis from each instrument are the strongest results of this manuscript. The authors convincingly demonstrate that the EESI is measuring ambient organic aerosol and that the variability captured by the factor analysis is from different primary and secondary sources. These results (along with the companion wintertime paper) lay important groundwork for establishing how EESI can be used to understand atmospheric aerosol composition and chemistry and will be very useful for future EESI measurements of ambient aerosol.

My main concern with the manuscript is that many of the authors' attributions of EESI signals to particular chemical processes are not strongly supported. For example, at several points the authors attribute components of SOA to either monoterpene, sesquiterpene, or aromatic VOC oxidation products but do not show any data from the co-located PTR-MS to support their arguments. Similarly, the authors simultaneously argue that the SOA being measured is representative of high-NOx conditions (p 11 ln 30) while also attributing SOA constituents to HOMs produced through low-NOx autoxidation (p 18 ln 35). At a minimum the authors should report the NO concentrations from the NABEL site to support these arguments. In my opinion this paper is sitting at the borderline between being appropriate for Atmospheric Chemistry and Physics vs Atmospheric Measurement Techniques, and strengthening the connections between observed factors and specific chemical processes would add to the argument that this is in fact an ACP paper.

Specific comments:

P 2 ln 11: Cooking OA should be included in this list of primary OA sources. Several studies have shown substantial contributions of cooking emissions to POA or even total OA. (e.g., Hayes et al., 2013).

P 2 ln 15: Please consider a more recent reference to complement the Heald et al. reference. For example, see Shiravastava et al. 2017.
P 2 ln 26-30 & p 3 ln 3: I feel the value of the EESI time response relative to other state-of-the-art techniques is being overstated here. Currently this manuscript does not demonstrate the there is value in the 5-minute EESI data that could not be obtained at the 30-60 min time resolution of a FIGAERO or TAG measurement, and the authors do not acknowledge the time response of the CHARON here. Consider highlighting an event in this study where the OA chemical composition is varying on a 5-minute timescale or reworking this section.

P 2 ln 36 & p 3 ln 19: Equating the thermal decomposition of the CHARON and the AMS feels very unfair to the CHARON group. I strongly recommend that the authors separate the discussion of fragmentation in these instruments into different sentences that more accurately reflect the extent of thermal decomposition observed by each. Similarly, I encourage the authors to consider softening the claim on p 3 ln 19 that there is no thermal decomposition or ionization-induced fragmentation in the EESI. To my knowledge the full extent of the investigation into this is the ~10 standards reported in Lopez-Hilfiker et al. 2019, which only demonstrated that polyols and dicarboxylic acids do not fragment. The components of the SOA measured in the study, especially organonitrates, are likely to be considerably more fragile than those standards and so I am not convinced that there is sufficient evidence to support this claim.

P 5 ln 22: The measurement of nicotine as [M+H]+ makes me doubt that other ionization pathways are "almost entirely suppresse[d]". It seems very likely that many amines are sufficiently basic to also be detected through a protonation pathway.

P 12 ln 31: As discussed above - authors mention that there were PTR-MS and NOx measurements at the sampling site but do not show any results. Is there any evidence from those measurements that would support the assignment of EESI SOA factors to NO3-monoterpene reactions? Or is the attribution solely based on assigned elemental compositions and the diurnal profile?

P 15 ln 21: The authors should report a bulk sensitivity to OA during this study, calculated from the slope of the regression in Fig. 10a and compare it to the bulk SOA EESI sensitivities reported in Lopez-Hilfiker et al. 2019. Much of the discussion in this manuscript is heavily influenced by the fact that the total EESI signal shows decent correlation to AMS OA mass, despite significant variability in compound-to-compound sensitivities (and the inability to measure HOA). Being able to compare the EESI's bulk OA sensitivity to single-compound sensitivities & single-precursor SOA sensitivities would help the reader understand the extent to which the correlation with the AMS measurement may be driven by a handful of high-sensitivity compounds that correlate with total OA. This is especially important because the authors do not show any EESI calibration data.

P 16 ln 38: This paragraph is very confusing and I do not follow what readers are supposed to conclude about how extraction in EESI is affecting bulk molecular properties as compared to the AMS.

P 17 ln 16: How confidently can you conclude that having a single factor above 50% of EESI signal indicates that the factor is >50% of OA mass? If the EESI is twice as sensitive to SOA as COA, and totally blind to HOA, it seems like an SOA factor could have higher sensitivity than bulk SOA and dominate signal while being a small fraction of OA mass.

P 18 ln 29: In this paragraph the claims that ions with an assigned elemental composition correspond to the same molecules measured in other studies need additional support. Based on the number of compounds detected in offline electrospray aerosol measurements overlapping assigned formulas are expected and don't necessarily point to a shared source. Is there evidence in the NO data from the Zurich site that HOM formation could have been occurring? Can this be reconciled with the claim discussed above that the chemistry forming the regional SOA is high-NOx?

Figure 10: In panel B the regression lines are thin, hidden behind the markers, and quite dark and monochromatic (to this reader's eyes). Consider some combination of

enlarging the figure, bringing the regression lines to the front, and making the colors easier to differentiate.

Minor comments:

P 5 ln 13: Please state the resolution achieved in this study instead of the nominal Tofwerk maximum.

P 5 ln 29: Was it the heated capillary or the electrospray capillary that clogged? Please clarify. Also please clarify what is meant by "dirty solution".

P 5 ln 33 & p 19 ln 19: clarify that "signal stability within $\pm$ 7.3%" is a relative standard deviation and not the range of signal intensity observed.

P 6 ln 7: In point #3 do you mean to say "subtracted from the ambient spectrum"?

P 11 ln 5: Close parentheses on the reference.

P 11 ln 9: Report nicotine as C10H14N2 to stay consistent with the omission of "Na+" from other assigned elemental compositions.

P 12 ln 3: This sentence is either missing a clause or the word "that" should be removed.

P 14 ln 1: Section 3.2.5 could be moved to SI. Good additional proof of the robustness of the factor analysis, but quite long and of less general interest than the rest of the manuscript.

P 18 ln 36 & p 19 ln 2, possibly others: elemental compositions with "0" instead of "O".

References:

Hayes, P. L., et al. (2013), Organic aerosol composition and sources in Pasadena, California during the 2010CalNex campaign, J. Geophys. Res. Atmos.,118, 9233–9257, doi:10.1002/jgrd.50530.

Shrivastava, M., et al. (2017), Recent advances in understanding secondary organic

aerosol: Implications for global climate forcing, Reviews of Geophysics, 55(2), 505-559.

Interactive
comment

---

## Author Comment (AC1) · 28 Oct 2019

**Response to reviewer's comments:**

**Title:** Organic aerosol source apportionment in Zurich using extractive electrospray ionization time-of-flight mass spectrometry (EESI-TOF): Part I, biogenic influences and day/night chemistry in summer.

**Journal:** Atmospheric Chemistry and Physics

**Manuscript ID:** acp-2019-361

Dear Editor,

We thank you and the reviewers for their comments. Our detailed point-by-point responses to the reviewers' comments (in black regular typeset) are provided in the blue regular typeset. The newly inserted/ revised text (highlighted in the main text) can be found in *grey italic typeset*.

**Reviewer 1:**

**Summary:**

This paper presents the first field observations using the EESI-TOF, a new online ESIMS instrument for aerosols. Factorisation is applied to the data and is compared against the equivalent factorisation products from an AMS. While the prevalence of cigarette smoke would throw into question the representativeness of the measurement site, it provides an interesting insight into the instrument function. Also, the separation of four different SOA factors adds an additional dimension to the AMS analysis, which typically only breaks it down into two factors. While this paper has a technical theme, I would consider there to be enough atmospheric science to warrant publication in ACP rather than AMT. It is also very well written and presented. This work will no doubt form the foundation for many future papers using this technique, much as the Lanz et al. papers did for PMF-AMS using data obtained in Zurich. I recommend publication subject to minor comments, however I only have a few of these to make.

**Comments:**

Page 5, line 8: It should be specified that the resolution being reported is m/dm. Also, this quantity is normally presented dimensionless; Th/Th isn't really a unit.

We agree and modify the unit reported as $m/\Delta m$, and the text as follows:

*"The final input matrix contained 281 ions (excluding isotopes and $CO_2$-dependent ions and 285 ions including the CO2-dependent ions) between m/z 12 and 120 at a resolution of 3000-4000 $m/\Delta m$, and 22182 points in time (with steps of 60 s)."*

Page 10, line 6: Why is the average of the bootstrap runs used as a solution? Bootstrapping is normally used to investigate the robustness of a solution, not to provide the solution itself. While the process is explained in greater detail in section 3.2.5, I still don't see why the average should be considered a more reliable solution than the base case (if it was because of the constraint on the cooking factor, couldn't the optimal a-value simply be applied to the base case?). Regardless, I would

have liked to have seen the technical explanation in 3.2.5 given before the presentation of results.

As discussed in the bootstrap analysis section, we do not obtain a single "best" solution or a single optimized a-value, but rather a set of acceptable solutions (obtained from 1000 bootstrap runs in which the $COA_{EESI}$ a-value is randomly selected in the range 0 to 1 with 0.1 step size). Therefore, the base case represents a single, quasi-randomly selected solution out of a large set of acceptable solutions rather than the "best" solution of this set. We therefore consider the average of all acceptable solutions to be the best representation of the source apportionment analysis (together with the associated uncertainties).

We appreciate the need to have sufficient discussion of the technical side of the analysis process prior to presentation of results. However, in this particular case, discussion of the bootstrap analysis requires the reader to already be familiar with specific details of the EESI-TOF PMF solution, in particular the characteristics of the different day and night $OOA_{EESI}$ factors. Therefore we prefer to leave the section in its original location. To address the reviewer's concern, we have added a brief technical summary of the bootstrap analysis, together with a summary of our reasons for focusing discussion on the averaged bootstrap solution rather than the base case, to the beginning of Section 3.2.1:

*"The 6-factor solution presented throughout the text is the averaged solution among 795 accepted bootstrap runs (out of 1000 total). The bootstrap analysis is discussed in detail in Section 3.2.5, and is based around random selection of the a-value constraints on the profile of a cooking-related factor ($COA_{EESI}$) (with a-values selected in the range 0 to 0.1, with a step size of 0.1), with the $COA_{EESI}$ anchor profile constrained using the cleaner cooking-related factor profile retrieved in the 7-factor solution. Critically, we consider all solutions classified as being reasonable and unmixed (according to the evaluation in Section 3.2.5) to be of equal merit. Therefore the base case solution therefore represents only a single, quasi-randomly-selected solution out of this large set (rather than an optimized solution) and we consider the average of all acceptable solutions to be the best representation of the source apportionment analysis."*

Page 10, line 26: How can you say that there were negligible local influences? The primary factors alone would indicate that local sources were very significant.

This statement was intended (1) to refer to the SOA fraction only and (2) to set up the contrast between the diurnal behavior of the total mass (flat) and the individual factors (strong patterns). We have clarified the text as follows.

*"The pattern of the sum of all SOA factors is basically flat. However, each individual SOA factor exhibits strong and distinctive variation."*

Page 12, line 20: Please refrain from using the term 'significant' in a statistical context unless a particular test (e.g. p-test) has been applied.

We adjusted the text removing the word "significant" where not referring to a statistical significance:

*"The time series $DaySOA1_{EESI}$ shows a correlation with $MO\text{-}OOA_{AMS}$ (R=0.54) which typically represents less volatile and more aged/regional, secondary organic aerosol compounds."*

**Reviewer 2:**

**Summary:**

This manuscript presents source apportionment from ambient measurements where an extractive electrospray (EESI) mass spectrometer was deployed alongside an aerosol mass spectrometer (AMS) in an urban setting. The paper demonstrates the capability of the EESI to measure variability in ambient organic aerosol that is consistent with that measured by the AMS, and to also identify additional variability in SOA composition that is not isolated in factor analysis of the AMS data. The comparison of the EESI and AMS total signal and the comparison of the factor analysis from each instrument are the strongest results of this manuscript. The authors convincingly demonstrate that the EESI is measuring ambient organic aerosol and that the variability captured by the factor analysis is from different primary and secondary sources. These results (along with the companion wintertime paper) lay important groundwork for establishing how EESI can be used to understand atmospheric aerosol composition and chemistry and will be very useful for future EESI measurements of ambient aerosol.

**General comments:**

My main concern with the manuscript is that many of the authors' attributions of EESI signals to particular chemical processes are not strongly supported. For example, at several points the authors attribute components of SOA to either monoterpene, sesquiterpene, or aromatic VOC oxidation products but do not show any data from the co-located PTR-MS to support their arguments. Similarly, the authors simultaneously argue that the SOA being measured is representative of high-NOx conditions (p 11 ln 30) while also attributing SOA constituents to HOMs produced through low-NOx autoxidation (p 18 ln 35). At a minimum the authors should report the NO concentrations from the NABEL site to support these arguments. In my opinion this paper is sitting at the borderline between being appropriate for Atmospheric Chemistry and Physics vs Atmospheric Measurement Techniques, and strengthening the connections between observed factors and specific chemical processes would add to the argument that this is in fact an ACP paper.

The reviewer raises several issues here: (1) attribution of SOA components to different precursor classes; (2) utility of the PTR-MS data for such attribution; (3) consistency of the discussion of $NO_x$ effects on SOA composition; and (4) NO concentrations at the NABEL station. We discuss these issues separately below.

(1) The assignments of the species have been supported by similarity with spectra of monoterpenes measured with the EESI-TOF system after direct injection and aging in one of our simulation chambers (Pospisilova et al., 2019). Furthermore, the molecular identity and factor profiles have been compared with mass spectra of gas-phase highly oxygenated molecules (HOMs) measured in the Finnish boreal forest with a $NO_3$-CIMS (Yan et al., 2016).

(2) We believe that the PTR-MS measurements would not be significantly helpful for this analysis. Specifically, the lifetime of monoterpenes is so short that there would not be present any straightforward relationship between PTR monoterpenes and monoterpene SOA. The clearest relationship would be total emitted monoterpenes vs. generated SOA and that would represent the exponential temperature relationship showed in Figure 6. As we anyhow did not use the PTR data we deleted the mentioning of the PTR in the manuscript

(3) The reviewer identifies two locations in the manuscript, where we agree the discussion of $NO_x$ effects on SOA composition should be clarified.

The first instance (p 11 ln 30) refers to the effects of $NO_x$ on dimer formation. Several studies (including those cited) indicate that $NO_x$ suppresses dimer formation, consistent with reduced dimer concentrations in Zurich vs. Hyyttiala (at comparable mass concentrations). We now also cite Pospisilova et al. (2019), where $\alpha$-pinene + $O_3$ in a $NO_x$-free simulation chamber is shown to yield increased dimer concentrations relative to Hyyttiala (again at comparable mass concentrations). Our use of "high $NO_x$" in the original manuscript was meant to discuss this gradient rather than providing a binary value between high/low $NO_x$ regimes, and the text has been revised to clarify this as follows:

*"However, the absence of signal from $C_{19}$ and $C_{20}$ compounds suggests that dimer concentrations are low in Zurich. This may be due to suppression of dimerization by $NO_x$ (Yan et al., 2016; Kurten et al., 2016), and is consistent with the dimer fraction here being low compared to that observed in the Finnish boreal forest, and with both ambient measurements being lower than that of $\alpha$-pinene ozonolysis in a $NO_x$-free simulation chamber (Pospisilova et al., 2019)."*

The second instance (p 18 ln 35) refers to a comparison of the molecular formulas identified in the current study with those from source apportionment of gas-phase data by a $NO_3$-CIMS in the Finnish boreal forest (Yan et al., 2016). Here, the main point, as also shown by Yan et al., is that nighttime and daytime chemistry are significantly different, also concerning the role of NO. We note that despite comparing gas and particle instruments, several ions with a common molecular formula are identified in the two studies and also have similar temporal behavior. Thus we can use the better-understood gas-phase chemistry to infer the major processes affecting the particle phase. This point is clarified in the text as follows:

*"Despite of compositional differences between the gas and particle phase, several ions having common molecular formulae are identified in both studies and have also similar temporal behavior. We use these correlations together with the better-understood gas-phase chemistry giving rise to the chosen ions to infer the major processes affecting the particle phase."*

(4) The NO concentration ranged between 0 and 44 µg/m$^3$ for the entire measurement period with an average value of 3 µg/m$^3$ (NABEL station Zurich-Kaserne measurements).

P 2 ln 11: Cooking OA should be included in this list of primary OA sources. Several studies have shown substantial contributions of cooking emissions to POA or even total OA. (e.g., Hayes et al., 2013).

We agree with the statement and adjusted the text as follows:

*"POA emissions typically include combustion of fossil fuels, direct injection of unburnt fuel and lubricants, industrial emissions, plant matter debris, biomass burning, cooking emissions and biogenic emissions (DeGouw et al., 2009, Hayes et al., 2013)."*

P 2 ln 15: Please consider a more recent reference to complement the Heald et al. reference. For example, see Shiravastava et al. 2017.

Agreed, the more recent reference has been added in the text following your suggestion:

*"However our capability to characterize SOA is limited (Heald et al., 2008, Shiravastava et al., 2017)."*

P 2 ln 26-30 & p 3 ln 3: I feel the value of the EESI time response relative to other state-of-the-art techniques is being overstated here. Currently this manuscript does not demonstrate the there is value in the 5-minute EESI data that could not be obtained at the 30-60 min time resolution of a FIGAERO or TAG measurement, and the authors do not acknowledge the time response of the CHARON here. Consider highlighting an event in this study where the OA chemical composition is varying on a 5-minute timescale or reworking this section.

For the current study, the main advantage of the EESI-TOF relative to a semi-continuous system such as the FIGAERO is the more accurate measurement of reactive species. The FIGAERO leaves the aerosol on a the filter for a long time before analysis ("long" being relative to the fastest decay rates for some oxidation products as established in Pospisilova et al., 2019). Such reactions (as well as reactions of collected OA with atmospheric oxidants on the filter surface) can potentially result in biases or errors in the measurement of OA composition even if the characteristic timescales for atmospheric composition changes are slower. Further, although not exploited in the current study the fast time response of the EESI-TOF is very important for mobile sampling, as demonstrated in Lopez-Hilfiker et al. (2019) explicitly presenting the time response values.

We agree with the reviewer that the CHARON system is capable of highly time-resolved measurements. However, the CHARON is omitted from this discussion in the paper because it is not comparable to the EESI or FIGAERO in terms of chemical resolution, because the proton-transfer reaction involves relatively high energy and SOA molecules fragment extensively (Muller et al., 2017).

P 2 ln 36 & p 3 ln 19: Equating the thermal decomposition of the CHARON and the AMS feels very unfair to the CHARON group. I strongly recommend that the authors separate the discussion of fragmentation in these instruments into different sentences that more accurately reflect the extent of thermal decomposition observed by each. Similarly, I encourage the authors to consider softening the claim on p 3 ln 19 that there is no thermal decomposition or ionization-induced fragmentation in the EESI. To my knowledge the full extent of the investigation into this is the ~10 standards reported in Lopez-Hilfiker et al. 2019, which only demonstrated that polyols and dicarboxylic acids do not fragment. The components of the SOA measured in the study, especially organonitrates, are likely to be considerably more fragile than those standards and so I am not convinced that there is sufficient evidence to support this claim.

The capabilities of the CHARON were misstated in the original manuscript; the issue is not with thermal decomposition (which is believed to be negligible) but rather with ionization-induced fragmentation, which Muller et al. (2017) showed to be dominant for SOA-like molecules. This has been clarified as follows:

*"For instance, the Aerodyne aerosol mass spectrometer (AMS) and the CHARON-PTR-ToF-MS are both able to describe bulk compositional properties of OA. However, the AMS subjects OA molecules to significant thermal decomposition and ionization-induced fragmentation. While thermal decomposition does not significantly influence the CHARON, the proton transfer reaction is of sufficiently high energy that molecules of the type found in SOA undergo significant fragmentation, with the signal from the parent ion comprising a very small fraction of the total (Müller et al., 2017)."*

Verifying a lack of thermal decomposition in complex aerosol is of course extremely difficult. As noted by the reviewer, Lopez-Hilfiker et al. (2019) showed that any aerosol with thermal stability greater or equal to that of citric acid (which decomposes at approximately 200 °C) should be preserved. As noted in the public discussion associated with Lopez-Hilfiker et al., we have confirmed that changing the capillary temperature (in either direction) does not affect the observed relative composition of α-pinene SOA and that changes in absolute sensitivity are consistent with changes in droplet evaporation efficiency rather than thermal decomposition. Finally, we note that thermally fragile compounds are observed in the EESI-TOF at high intensity: organonitrates in the current study, and dihydroxy- and trihydroxynitrates in laboratory experiments (Liu et al., 2019). This would be unexpected if thermal decomposition were significant. Taken together, we believe this evidence supports our description of the instrument capabilities and we therefore retain the original text.

P 5 ln 22: The measurement of nicotine as [M+H]+ makes me doubt that other ionization pathways are "almost entirely suppressed". It seems very likely that many amines are sufficiently basic to also be detected through a protonation pathway.

This is entirely possible, but in this analysis nicotine is the only compound detected as an adduct with $H^+$ instead of $Na^+$ and in the absence of direct observation we prefer not to speculate here.

P 12 ln 31: As discussed above authors mention that there were PTR-MS and NOx measurements at the sampling site but do not show any results. Is there any evidence from those measurements that would support the assignment of EESI SOA factors to NO3-monoterpene reactions? Or is the attribution solely based on assigned elemental compositions and the diurnal profile?

As outlined above we deleted the PTR-MS from the instrument list. Also, neither the PTR-MS nor the $NO_x$ measurements would provide an $NO_3$ concentration. Our assignments are confirmed by data from Hyytiala and chamber experiments, where the latter were done without $NO_3$ radicals. We mention here the possibility that organonitrates are likely derived from nitrate ($NO_3$) radical oxidation of monoterpenes at night, however, we do not yet have chamber experiments with $NO_3$ radicals to show here, so this remains a hypothesis.

P 15 ln 21: The authors should report a bulk sensitivity to OA during this study, calculated from the slope of the regression in Fig. 10a and compare it to the bulk SOA EESI sensitivities reported in Lopez-Hilfiker et al. 2019. Much of the discussion in this manuscript is heavily influenced by the fact that the total EESI signal shows decent correlation to AMS OA mass, despite significant variability in compound-to-compound sensitivities (and the inability to measure HOA). Being able to compare the EESI's bulk OA sensitivity to single-compound sensitivities & single-precursor SOA sensitivities would help the reader understand the extent to which the correlation with the AMS measurement may be driven by a handful of high-sensitivity compounds that correlate with total OA. This is especially important because the authors do not show any EESI calibration data.

The slopes in Fig. 10a and b, currently reported in EESI-TOF ag s$^{-1}$ per AMS µg m$^{-3}$, can be readily converted to EESI-TOF sensitivity (ions molec$^{-1}$) using the EESI-TOF flowrate of 1 L min$^{-1}$ (see Lopez-Hilfiker et al., 2019, Eqs. 1-3). Applying this to Fig. 10b, we estimate bulk sensitivities of 3.2 x 10$^{-8}$ ions molec$^{-1}$ to SOA, 2.8 x 10$^{-8}$ ions molec$^{-1}$ to CS-OA, and 9.0 x 10$^{-9}$ ions molec$^{-1}$ to COA. An overall bulk sensitivity to OA can also be estimated from Fig. 10a (1.9 x 10$^{-8}$), however, as noted by the

reviewer, this value is somewhat less meaningful because it includes the HOA component that is not detected (likely not extracted/ionized) by the EESI-TOF. These values have been added to the figure caption. The measurements performed by Lopez-Hilfiker et al. (2019) were conducted in the laboratory, meaning that mass spectrometer tuning, electrospray voltages, positioning, and flowrate, ESI capillary state of wear, and other operational factors differ from the current campaign. In our experience, these factors would likely govern the inter-campaign comparison suggested by the reviewer, and would not allow any conclusions to be drawn regarding the relative response of the EESI-TOF to different aerosol composition and so we do not include this information in the manuscript. Nevertheless, to answer the reviewer's question we note here that in the simulation chamber experiment reported in Fig. 10 of Lopez-Hilfiker et al. (2019), a bulk sensitivity of $3.0 \times 10^{-8}$ ions molec$^{-1}$ was observed for SOA generated from the ozonolysis of α-pinene, which is comparable to the values obtained here.

The reviewer's second point regards the potential of a few high-sensitivity compounds to dominate the overall apportionment. This could indeed be an issue for the CS-OA factor, where nicotine and $C_6H_{10}O_5$ alone provide ~20% of the signal. However, as shown in Fig. 2, for all other factors the signal is instead divided across a large set of ions, meaning any small subset of ions contains an insufficient fraction of the signal to independently drive the temporal behavior.

P 16 ln 38: This paragraph is very confusing and I do not follow what readers are supposed to conclude about how extraction in EESI is affecting bulk molecular properties as compared to the AMS.

Here we investigate one possible hypothesis to explain the observed differences in bulk molecular composition between the AMS and EESI-TOF, namely that the difference arrives from reduced extraction efficiency of less soluble compounds into the EESI spray. We assess this by comparing AMS measurements of ambient aerosol to offline AMS measurements of re-nebulized water extracts. If extraction were the key difference between AMS and EESI, we would expect the ambient EESI data to fall between the AMS ambient and offline AMS water extracts. However, this is not the case, and the online and offline AMS data are more similar to each other in elemental ratio than either of them to the EESI data. Therefore, we suggested here that probably the discrepancy in elemental ratio between the online AMS and EESI data is not solely driven by the extraction technique, however, without drawing any further conclusions. The text has been revised for clarity, as follows:

*"Because the first step in EESI-TOF detection is a rapid extraction into the methanol/water droplets generated by the electrospray, one possibility for the observed discrepancies in the O:C ratios between the AMS and the EESI-TOF could be incomplete extraction of less soluble components in the EESI-TOF. To investigate this, we compare the O:C ratios from the AMS factors retrieved in the current study (COA$_{AMS}$ 0.1, HOA$_{AMS}$ 0.057 and OOA$_{AMS}$ 0.42-0.5) with those from offline AMS source apportionment of aqueous filter extracts, where water-insoluble components are not detected. The offline-AMS method yields O:C ratios consistent with the online AMS (COA$_{offline}$ 0.10, HOA$_{offline}$ 0.06, and OOA$_{offline}$ 0.51) (Bozzetti et al., 2017). In contrast, the EESI-TOF ratios are significantly higher (COA$_{EESI}$ 0.38 and SOA$_{EESI}$ 0.56-0.62) despite of extraction into a water/methanol mixture rather than water-only. This suggests that the EESI extraction process (i.e. solubility) alone cannot explain the discrepancies between the two instruments. Note that this assumes no kinetic limitations on solubility/extraction, as the offline method applies a water extraction for 20 min, while the EESI-TOF uses a very fast extraction in water/methanol; however, this assumption is likely valid as laboratory tests suggest complete extraction of particles by the EESI-TOF in the measured size range (Lopez-Hilfiker et al., 2019)."*

P 17 ln 16: How confidently can you conclude that having a single factor above 50% of EESI signal indicates that the factor is >50% of OA mass? If the EESI is twice as sensitive to SOA as COA, and totally blind to HOA, it seems like an SOA factor could have higher sensitivity than bulk SOA and dominate signal while being a small fraction of OA mass.

We agree in principle, and this was a motivating factor behind the detailed comparison of the AMS and EESI-TOF source apportionment results presented in Section 3.3. We note in particular that the sum of the EESI-TOF SOA factors and the sum of the AMS OOA factors are in excellent agreement, as shown in Fig. 4 (bottom panel). Such agreement would not be possible unless the relative sensitivities to the four EESI-TOF SOA factors are similar. Therefore, while we do not propose a precise 1:1 relationship between the mass fraction of a factor measured by the EESI-TOF and its actual ambient mass fraction, a mass fraction >50% certainly indicates a major contribution to the overall composition. This is clarified in the text as follows:

*"The nighttime composition is significantly different, with $NightSOA2_{EESI}$ in particular often being at or above 50% of the total SOA as measured by the EESI-TOF while the AMS analysis does not allow identification of this factor. We note that according to Fig. 4 (bottom panel) the total AMS OOA is well-correlated with the sum of all four EESI-TOF SOA factors, indicating that the high contribution of $NightSOA2_{EESI}$ reflects a large contribution to the total SOA mass rather than an anomalously high relative sensitivity in the EESI-TOF."*

P 18 ln 29: In this paragraph the claims that ions with an assigned elemental composition correspond to the same molecules measured in other studies need additional support. Based on the number of compounds detected in offline electrospray aerosol measurements overlapping assigned formulas are expected and don't necessarily point to a shared source. Is there evidence in the NO data from the Zurich site that HOM formation could have been occurring? Can this be reconciled with the claim discussed above that the chemistry forming the regional SOA is high-NOx?

This point was discussed in detail above (point 3) of the reply to comments #1 by Reviewer 2), and our response is repeated here:

The first instance (p 11 ln 30) refers to the effects of NOx on dimer formation. Several studies (including those cited) indicate that NOx suppresses dimer formation, consistent with reduced dimers in Zurich vs. Hyyttiala (at comparable mass concentrations). We now also cite Pospisilova et al. (2019), where $\alpha$-pinene + $O_3$ in a $NO_x$-free simulation chamber is shown to yield increased dimer concentrations relative to Hyyttiala (again at comparable mass concentrations). Our use of "high NOx" in the original manuscript was meant to discuss this gradient rather than providing a binary value between high/low NOx regimes, and the text has been revised to clarify this as follows:

*"However, the absence of signal from $C_{19}$ and $C_{20}$ compounds suggests that dimer concentrations are low in Zurich. This may be due to suppression of dimerization by $NO_x$ (Yan et al., 2016; Kurten et al., 2016), and is consistent with the dimer fraction here being low compared to that observed in the Finnish boreal forest, and with both ambient measurements lower than that of $\alpha$-pinene ozonolysis in a $NO_x$-free smog chamber (Pospisilova et al., 2019)."*

Figure 10: In panel B the regression lines are thin, hidden behind the markers, and quite dark and monochromatic (to this reader's eyes). Consider some combination of enlarging the figure, bringing the regression lines to the front, and making the colors easier to differentiate.

We agree and therefore we modify Figure 10 panel B as suggested:

[Figure]

We also modified the Figure 10 panel A, forcing the intercept to 0 and obtaining a more representative fit for the data not influenced by outliers:

[Figure]

**Specific comments:**

P 5 ln 13: Please state the resolution achieved in this study instead of the nominal Tofwerk maximum.

The resolution ($m/\Delta m$) is now reported in the text as follows:

*"The final input matrix contained 281 ions (excluding isotopes and $CO_2$-dependent ions and 285 ions including the $CO_2$-dependent ions) between m/z 12 and 120 at a resolution of 3000-4000 $m/\Delta m$, and 22182 points in time (with steps of 60 s)."*

P 5 ln 29: Was it the heated capillary or the electrospray capillary that clogged? Please clarify. Also please clarify what is meant by "dirty solution".

The clogging occurred in the electrospray capillary and with "dirty solution" we refer to some extent of contamination, which decrease the purity of the solution. Clarification included in the text as follows:

*"The remaining ~15% loss of data acquisition was due to instrumental issues, e.g. clogged electrospray capillary resulting in loss of the signal or "dirty solution" to substitute (contamination from ambient air decreasing the purity of the solution)."*

P 5 ln 33 & p 19 ln 19: clarify that "signal stability within ± 7.3%" is a relative standard deviation and not the range of signal intensity observed.

We agree and we adjusted the sentences as following:

*"The (NaI)Na$^+$ signal, an approximate surrogate for ion source stability, varied by ± 7.3 % (relative standard deviation) across the entire campaign and exhibited no systematic drift (Fig. S2), and no corrections relating to sensitivity drift were applied."*

*"The EESI-TOF measured for 3 weeks during summer in Zurich, Switzerland, achieving >85% data coverage without any systematic drift and signal stability within ± 7.3 % (relative standard deviation). Overall, the campaign demonstrated the EESI-TOF to be a sufficiently robust instrument for field operation."*

P 6 ln 7: In point #3 do you mean to say "subtracted from the ambient spectrum"?

Yes, we apologize for the mistake and we adjusted the text as follows:

*"3) The estimated background was subtracted from the ambient spectrum and the resulting difference matrix re-averaged to 300 s time resolution for PMF analysis."*

P 11 ln 5: Close parentheses on the reference.

Yes, thank you. Text adjusted also updating the reference itself:

*"Further, the EESI-TOF is probably more sensitive to levoglucosan than to bulk SOA (Lopez-Hilfiker et al., 2019). During this study, it is likely emitted from open cooking activities in the vicinity of the measurement site."*

P 11 ln 9: Report nicotine as C10H14N2 to stay consistent with the omission of "Na+" from other assigned elemental compositions.

We agree. Text adjusted as following:

*"The CS-OA$_{EESI}$ mass spectrum is dominated by $C_{10}H_{14}N_2$ (nicotine, m/z 163.12), and $C_6H_{10}O_5$ (levoglucosan) which contribute 15% and 10%, respectively, to the profile signal."*

P 12 ln 3: This sentence is either missing a clause or the word "that" should be removed.

Sentence adjusted as follows:

*"However, fragmentation results in products progressively more difficult to distinguish from ring-opening products from the oxidation of aromatic precursors, and therefore we cannot rule out a contribution to these ions from aromatic oxidation products."*

P 14 ln 1: Section 3.2.5 could be moved to SI. Good additional proof of the robustness of the factor analysis, but quite long and of less general interest than the rest of the manuscript.

Considering the comment and suggestion from the RC1, who considers this section relevant and proposed to move it before the Results part we decided to move Section 3.2.5 above the Results section where it now appears as Section 3.2.1.

P 18 ln 36 & p 19 ln 2, possibly others: elemental compositions with "0" instead of "O".

Thank you for noticing this. Text adjusted as follows:

[revised manuscript text omitted]